# An increase of inhibition drives the developmental decorrelation of neural activity

**Mattia Chini[1]\*, Thomas Pfeffer[2], Ileana Hanganu-Opatz[1]**

[1]Institute of Developmental Neurophysiology, Center for Molecular Neurobiology, University Medical Center Hamburg-Eppendorf, Hamburg, Germany; [2]Center for Brain and Cognition, Computational Neuroscience Group, Universitat Pompeu Fabra, Barcelona, Spain

**Abstract** Throughout development, the brain transits from early highly synchronous activity patterns to a mature state with sparse and decorrelated neural activity, yet the mechanisms underlying this process are poorly understood. The developmental transition has important functional consequences, as the latter state is thought to allow for more efficient storage, retrieval, and processing of information. Here, we show that, in the mouse medial prefrontal cortex (mPFC), neural activity during the first two postnatal weeks decorrelates following specific spatial patterns. This process is accompanied by a concomitant tilting of excitation-inhibition (E-I) ratio toward inhibition. Using optogenetic manipulations and neural network modeling, we show that the two phenomena are mechanistically linked, and that a relative increase of inhibition drives the decorrelation of neural activity. Accordingly, in mice mimicking the etiology of neurodevelopmental disorders, subtle alterations in E-I ratio are associated with specific impairments in the correlational structure of spike trains. Finally, capitalizing on EEG data from newborn babies, we show that an analogous developmental transition takes place also in the human brain. Thus, changes in E-I ratio control the (de) correlation of neural activity and, by these means, its developmental imbalance might contribute to the pathogenesis of neurodevelopmental disorders.

**\*For correspondence:**
mattia.chini@zmnh.uni-hamburg.
de

**Competing interest:** The authors declare that no competing interests exist.

## Editor's evaluation

This manuscript presents a combination of in vivo recording and optogenetic experiments that together with modeling bring findings with important significance: inhibition is functionally present in the newborn frontal cortex having major effects on EEG dynamics. These important findings challenge the view on the switch in GABAergic excitation to inhibition and extend phenomenological observations to human infant EEG data. The strength of evidence is solid, with appropriate methodology used and only minor weaknesses noted regarding the human infant data.

## Introduction

Neural activity in the developing brain has several unique traits, such as discontinuity (*Chini and Hanganu-Opatz, 2021*), extremely low firing rates (*Shen and Colonnese, 2016*), a loose temporal coordination of excitation and inhibition (*Dorrn et al., 2010*), and weak modulation by behavioral state (*Cirelli and Tononi, 2015*; *Chini et al., 2019*). These patterns of early activity have been described in humans (*Vanhatalo and Kaila, 2006*) as well as in disparate model organisms ranging from fish (*Avitan et al., 2017*) to flies (*Akin et al., 2019*), and from rodents (*Khazipov et al., 2004*) to brain organoids (*Trujillo et al., 2019*). As brain networks mature, they gradually evolve into exhibiting motives

with adult-like spatiotemporal properties. Oscillatory events become more rhythmic, increasing their amplitude and average frequency (*Bitzenhofer et al., 2020*), oscillation patterns become more complex (*Trujillo et al., 2019*), the ratio of excitatory and inhibitory conductances (E-I ratio) decreases (*Zhang et al., 2011*), excitation and inhibition tighten on a temporal scale (*Dorrn et al., 2010*; *Moore et al., 2018*), and brain activity decorrelates and sparsifies (*Golshani et al., 2009*; *Rochefort et al., 2009*). The relationship between decorrelation and E-I ratio has been the subject of extensive experimental and theoretical work in the adult brain (*Zhou and Yu, 2014*; *Yu et al., 2014*; *Shew et al., 2014*). Decorrelated and sparse activity is a hallmark of adult spike trains (*Olshausen and Field, 2004*; *Vinje and Gallant, 2000*) and artificial neural networks alike (*Cun et al., 1990*; *Frankle and Carbin, 2019*; *Cogswell et al., 2016*). This activity pattern bears important functional and behavioral relevance, as it allows for efficient storing and retrieval of information, while minimizing energy consumption (*Olshausen and Field, 2004*; *Cun et al., 1990*; *Frankle and Carbin, 2019*). However, it is still unknown whether changes in E-I ratio underlie the developmental decorrelation of brain activity. The (patho)physiological relevance of this process is underscored by the hypothesis that altered E-I ratio is the hallmark of later-emerging neurodevelopmental disorders, such as autism (*Trakoshis et al., 2020*; *Antoine et al., 2019*; *Sohal and Rubenstein, 2019*; *Gao and Penzes, 2015*; *Medendorp et al., 2021*) and schizophrenia (*Gao and Penzes, 2015*; *Ferguson and Gao, 2018*), diseases that have also been linked to disruption of correlated activity in animal models (*Luongo et al., 2016*; *Hamm et al., 2017*; *Zick et al., 2018*).

E-I ratio is controlled by the interplay between pyramidal neurons (PYRs) and interneurons (INs). Throughout development, both populations of neurons migrate into the cortex following an 'inside-out' sequence that corresponds to their birthdate (*Lim et al., 2018*; *Sidman and Rakic, 1973*). This process is guided by cues provided by PYRs, which populate the cortical layers at an earlier time point than INs (*Lim et al., 2018*). The functional integration of INs into the cortical circuitry is a slow process that is initiated by the establishment of transient circuits. During the first postnatal week, mouse inhibitory circuits are dominated by somatostatin-positive (SST$^+$) INs (*Tuncdemir et al., 2016*; *Marques-Smith et al., 2016*). At later time points, parvalbumin-positive (PV$^+$) INs are also integrated into local networks (*Tuncdemir et al., 2016*; *Marques-Smith et al., 2016*; *Guan et al., 2018*). In rodents, the development of inhibitory synapses is not complete until postnatal day (P)30 (*Gour et al., 2021*). It is thus conceivable that the developmental strengthening of inhibitory synapses and the ensuing tilting of E-I ratio toward inhibition (*Zhang et al., 2011*) might underlie the decorrelation of neural activity. In favor of this hypothesis, chronic manipulation of IN activity in the murine barrel cortex results in altered temporal and spatial structure of brain activity (*Che et al., 2018*; *Duan et al., 2018*; *Modol et al., 2020*). However, direct evidence linking E-I ratio and the strengthening of inhibition with the developmental decorrelation of brain activity is still lacking. Further, it is still debated whether GABA, the main inhibitory neurotransmitter in the adult brain, actually exerts an inhibitory effect also during early development. It has long been thought that, in the rodent neocortex, GABA might act as an excitatory neurotransmitter for the first 2 postnatal weeks, and only subsequently 'switch' to an inhibitory effect (*Ben-Ari, 2002*; *Kalemaki et al., 2021*). These findings have been called into question (*Che et al., 2018*; *Kirmse et al., 2015*; *Murata and Colonnese, 2020*), but the argument remains open.

Here, we combined in vivo extracellular electrophysiological recordings and optogenetics with neural network modeling to systematically explore the relationship between E-I ratio and the decorrelation of neural activity in the developing rodent and human brain. In the murine medial prefrontal cortex (mPFC), a brain area where E-I ratio is of particular relevance in the context of neurodevelopmental disorders (*Trakoshis et al., 2020*), we show that GABA inhibits neural activity from the very first postnatal days, and that an increase in the strength of the exerted inhibition progressively decorrelates the prefrontal spike trains. Using neural network modeling and bidirectional optogenetic manipulation of IN activity, we further uncover how the inhibition increase drives the change in the correlation structure. Moreover, in a mouse model of impaired neurodevelopment, we report that excessively low E-I ratio causes impaired spike train correlations. Finally, we investigate two different EEG datasets and illustrate the translational relevance of these findings by providing first insights into analogous developmental processes that take place in newborn babies.

## Results

### The patterns of prefrontal activity dynamically evolve with age

To investigate the relationship between E-I ratio and the decorrelation of neural activity that occurs throughout development, we interrogated a large dataset (n=117 mice) of multi-site extracellular recordings of local field potential (LFP) and single unit activity (SUA) from the prelimbic subdivision of the mPFC of unanesthetized P2-12 mice (*Figure 1A*). Across this developmental phase, the LFP evolves from an almost complete lack of activity (silent periods) to uninterrupted (continuous) activity, passing through intermediate stages in which silent periods alternate with bouts of neuronal activity (active periods) (*Figure 1A*). To quantify this transition, we calculated the proportion of active periods over the recording and found that it monotonically increases over age (age slope = 0.88, 95% CI [0.84; 0.93], p<10^{-50}, generalized linear model) (*Figure 1B*). The increase in the proportion of active periods resulted from the augmentation of both number and duration of individual active periods until continuous activity was detected (*Figure 1—figure supplement 1A-C*). Accompanying these high-level changes in activity dynamics, the maximum amplitude of active periods, the broadband LFP power, and the SUA firing rate exponentially increased over age (age slope = 0.24, 0.47, and 0.21, 95% CI [0.18; 0.30], [0.40; 0.53], and [0.15; 0.27], p<10^{-9}, p<10^{-23}, p<10^{-11}, respectively, generalized linear model) (*Figure 1C*, *Figure 1—figure supplement 1D-F*).

Changes in the log-log power spectral density (PSD) slope (reflected by the 1/f exponent) have been linked to E-I ratio by several experimental (*Trakoshis et al., 2020*; *Lendner et al., 2020*; *Colombo et al., 2019*) and theoretical studies (*Trakoshis et al., 2020*; *Lombardi et al., 2017*; *Gao et al., 2017*). In particular, a relative increase in inhibition is thought of leading to a steeper PSD slope (higher 1/f exponent), whereas the opposite occurs when E-I ratio shifts toward excitation. Given that INs have a more protracted integration into cortical circuits than PYRs, this process might be accompanied by a developmental shift of the E-I ratio toward inhibition. In line with this hypothesis, the PSD slope grew steeper over age, as readily observed when the PSD was normalized by the area under the curve (*Figure 1D*). To quantify this observation, we parameterized the PSDs using a recently published protocol (*Donoghue et al., 2020*), and confirmed that the 1/f exponent increases with age (age slope = 0.12, 95% CI [0.11; 0.14], p<10^{-27}, linear model) (*Figure 1E*).

Thus, the monitoring of age-dependent dynamics of prefrontal LFP and SUA let us propose that, throughout development, E-I ratio tilts toward inhibition.

### E-I ratio controls pairwise spike train correlations in a neural network model

To explore the relationship between the 1/f exponent, E-I ratio, and the (de)correlation of neuronal spike trains, we simulated a neural network of 400 interconnected conductance-based leaky integrate-and-fire (LIF) neurons (*Figure 2A*). In line with anatomical data (*Markram et al., 2004*; *Hendry et al., 1987*), 80% of those simulated neurons were excitatory (PYRs), whereas 20% were inhibitory (INs). PYRs were simulated with outgoing excitatory AMPA synapses, while INs were simulated with outgoing inhibitory GABAergic synapses, including recurrent connections for both PYRs and INs. In keeping with theoretical and experimental work (*Buzsáki and Mizuseki, 2014*; *Hazan and Ziv, 2020*), excitatory and inhibitory synaptic weights were simulated with a lognormal distribution. Both neuron types received input noise and PYRs received an additional external excitatory Poisson stimulus with a constant spike rate of 1.5 spikes/s (see Materials and methods for details on the model). We parametrically varied the AMPA and GABA conductances on both PYRs and INs and defined the network's net inhibition strength as the ratio between the inhibitory and excitatory conductances. The network was simulated for 60 s for each parameter combination. The network's LFP was defined as the sum of the absolute values of all synaptic currents on PYRs, which was shown to be a reliable proxy of experimental LFP recordings (*Trakoshis et al., 2020*; *Mazzoni et al., 2008*). Across all parameters combinations, INs exhibited higher average firing rates compared to PYRs (INs = 4.06 Hz, 95% CI [3.53; 5.42], PYRs = 1.51 Hz, 95% CI [1.02; 2.98]).

In agreement with previous results (*Trakoshis et al., 2020*; *Gao et al., 2017*), the increase in the network's net inhibition strength robustly tilted the PSD slope (in the range from 30 to 100 Hz). Increased levels of net inhibition were associated with a steeper decay of power as a function of frequency and a corresponding increase of the 1/f exponent, across a range of AMPA conductance levels (*Figure 2B*); Pearson correlation coefficient, averaged across AMPA levels (inhibition strength

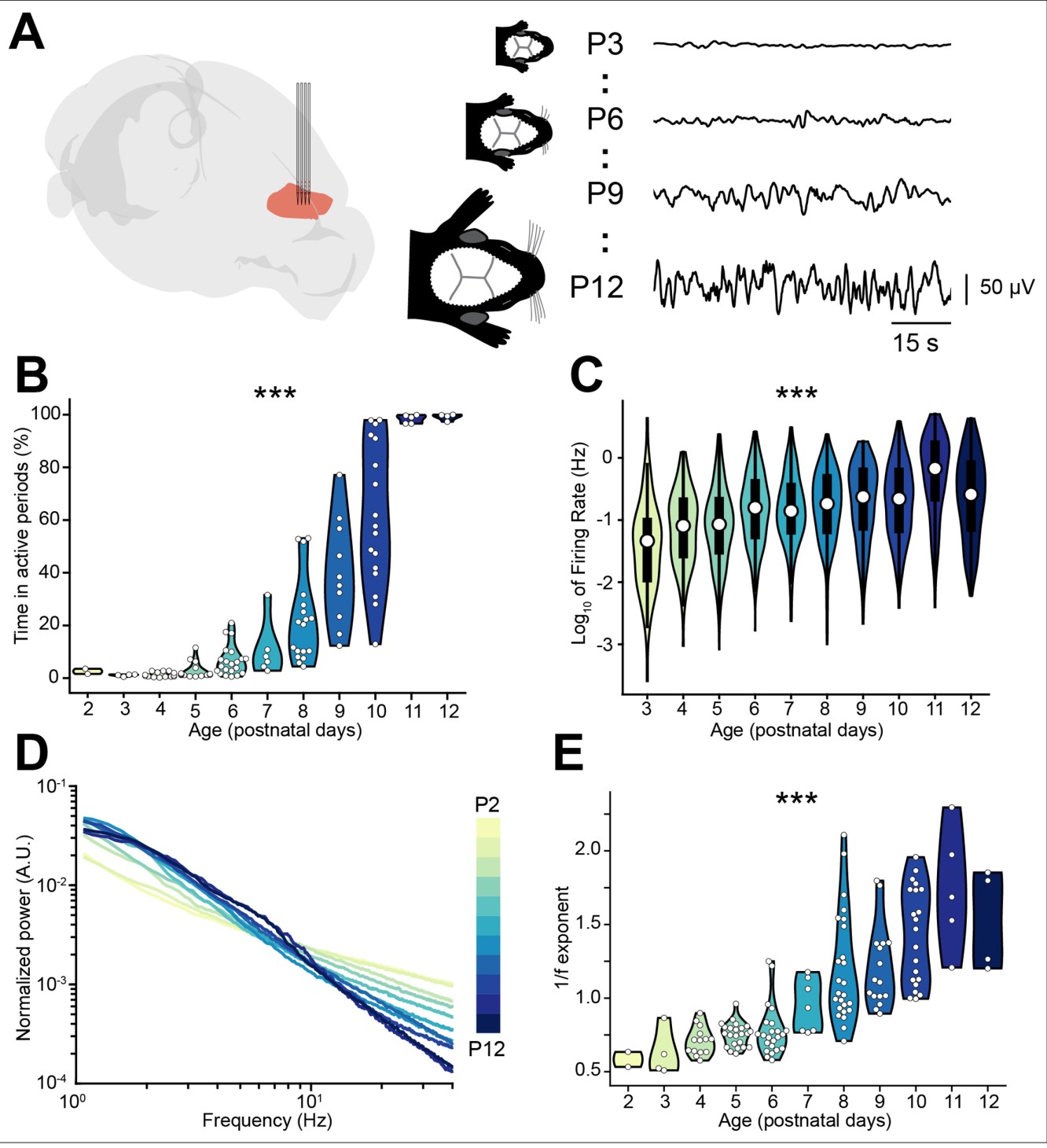

**Figure 1.** Active periods and local field potential (LFP) properties of the mouse medial prefrontal cortex (mPFC) across the first 2 postnatal weeks. (**A**) Schematic representation (*Claudi et al., 2021*) of extracellular recordings in the mPFC of P2-12 mice (left), and representative LFP traces from P3, P6, P9, and P12 mice (right). (**B and C**) Violin plots displaying the percentage of time spent in active periods (**B**) and the single unit activity (SUA) firing rate (**C**) of P2-12 mice (*n*=117 mice and 2269 single units, respectively). (**D**) Log-log plot displaying the normalized median power spectral density (PSD) power in the 1–40 Hz frequency range of P2-12 mice (*n*=117 mice). Color codes for age with 1 day increment. (**E**) Violin plot displaying the 1/f exponent of P2-12 mice (*n*=117 mice). In (**B**) and (**E**) white dots indicate individual data points. In (**C**) data are presented as median, 25th, 75th percentile, and interquartile range. In (**B**), (**C**), and (**E**) the shaded area represents the probability distribution density of the variable. In (**D**) data are presented as median. Asterisks in (**B**), (**C**), and (**E**) indicate significant effect of age. \*\*\*p<0.001. Generalized linear models (**B–C**) and linear model (**E**). For detailed statistical results, see *Supplementary file 1*.

*Figure 1 continued on next page*

**eLife** Research article

Neuroscience

slope = 0.022, 95% CI [0.021; 0.024], linear model). We next examined the effect of increasing the network's net inhibition strength on neural correlations. To this end, we computed the spike time tiling coefficient (STTC; at a lag of 1 s), a parameter that measures pairwise correlations between spike trains without being biased by firing rate (*Cutts and Eglen, 2014*), on the network's spike matrices (both PYRs and INs) and across all levels of net inhibition strength. For all levels of AMPA conductance, we found that increased net inhibition strength results in a robust logarithmic decrease in STTC (*Figure 2C*; inhibition strength slope = −0.068, 95% CI [0.075; 0.061], generalized linear model).

**Figure supplement 1.** Local field potential (LFP) and single unit activity (SUA) properties of the mouse medial prefrontal cortex (mPFC) across the first 2 postnatal weeks.

Thus, simulations of a biologically plausible neural circuit reveal that increased net inhibition strength leads to an increase of the PSD 1/f exponent that is accompanied by decorrelation of neural spike trains.

## Prefrontal spike trains decorrelate over development

Since neural network modeling predicts that a shift of E-I ratio toward inhibition leads to higher 1/f exponent and decorrelation of neural activity, we tested on the experimental data whether the developmental increase in the 1/f exponent in the mouse mPFC was accompanied by a decorrelation of neural activity. For this, we calculated the STTC between >40,000 pairs of spike trains over a large range of lags (2.5 ms to 10 s) (*Figure 3A*). For the analysis, we only considered SUA that was recorded for at least 60 min. To verify the robustness of STTC as an estimator, we compared the STTC obtained on the first and the second half of the recording. The STTCs computed on the two halves of the recording strongly correlated with each other across all the investigated lags (0.70, [0.50; 0.80] median and min-max Pearson correlation; 0.70 [0.52; 0.79] median and min-max Spearman correlation) (*Figure 3—figure supplement 1A-B*), thus corroborating its robustness as an estimator. Throughout the manuscript, we will consider STTC computed at 1 s, yet the summary plots and the supplementary statistical table include values calculated at all lags.

Using multivariate mixed hierarchical linear regression, we found that STTC negatively correlated with the distance between neurons (i.e. nearby neurons had higher STTC values than neurons that are far apart) over all the investigated lags (main distance effect, $p<10^{-78}$ at 1 s lag, linear mixed-effect model) (*Figure 3B–D*). This is in line with previous studies conducted in the adult (*Smith and Kohn, 2008*; *Goltstein et al., 2015*; *Greenberg et al., 2008*) and developing brain (*Golshani et al., 2009*; *Cutts and Eglen, 2014*; *Blankenship et al., 2011*) of several mammalian species. Further, STTC values negatively correlated with age at lags ≤5 s (main age effect, $p<10^{-4}$ at 1 s lag, linear mixed-effect

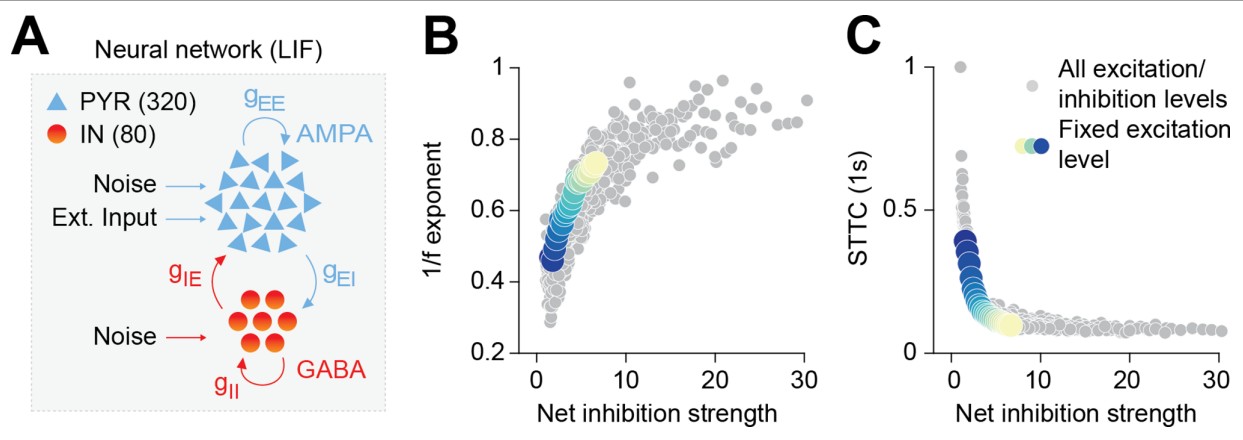

**Figure 2.** Increased inhibition leads to an increase in the 1/f exponent and decorrelates spike trains in a neural network model. (**A**) Schematic representation of the neural network model. (**B**) Scatter plot displaying the 1/f exponent as a function of net inhibition strength. (**C**) Scatter plot displaying average STTC as a function of net inhibition strength. For (**B**) and (**C**) color codes for inhibition strength with fixed excitation level.

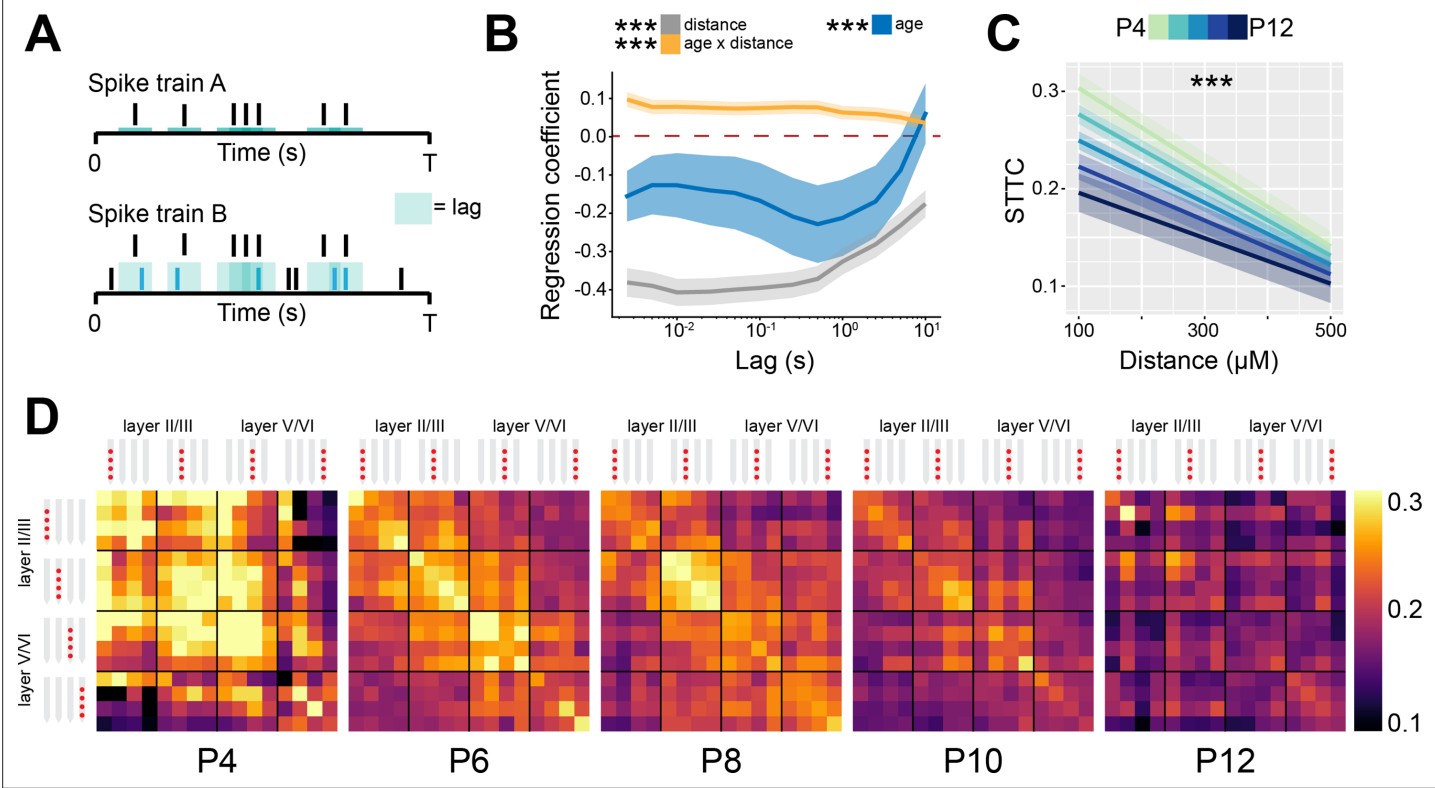

**Figure 3.** Spike time tiling coefficient (STTC) decreases throughout development with a specific spatial profile in the mouse medial prefrontal cortex (mPFC). (**A**) Schematic representation of the STTC quantification. (**B**) Multivariate linear regression coefficients with respect to STTC lag (n=40,921 spike train pairs and 82 mice). (**C**) Average STTC at 1 s lag of P4, P6, P8, P10, and P12 mice over distance (n=40,921 spike train pairs and 82 mice). Color codes for age. (**D**) Weighted adjacency matrices displaying average STTC at 1 s lag of P4, P6, P8, P10, and P12 mice as a function of the recording sites in which the spike train pair has been recorded. Color codes for STTC value. In (**B**) regression coefficients are presented as mean and 95% CI. In (**C**) data are presented as mean ± SEM. Asterisks in (**B**) indicate significant regression coefficients of the respective (interaction between) variables for STTC at 1 s lag. Asterisks in (**C**) indicate significant effect of age*distance interaction. ***p<0.001. Linear mixed-effect models. For detailed statistical results, see *Supplementary file 1*.

The online version of this article includes the following figure supplement(s) for figure 3:

**Figure supplement 1.** The spike time tiling coefficient (STTC) developmental decrease follows specific spatial patterns in the mouse medial prefrontal cortex (mPFC).

model), an effect that was strongest in the 100–1000 ms range (*Figure 3B and D*, *Figure 3—figure supplement 1C*). This developmental STTC decrease did not occur uniformly across all neuron pairs. Rather, age and distance had a significant interaction, nearby pairs of neurons displaying a more severe decorrelation over age than neurons that were further apart (age*distance interaction, p<10⁻¹² at 1 s lag, linear mixed-effect model) (*Figure 3B–D*).

Taken together, these data indicate that, throughout development, as E-I ratio tilts toward inhibition, there is a concomitant decorrelation of pairwise neuronal activity computed over lags that span more than three orders of magnitude. This result is in agreement with data from the rodent barrel cortex (*Golshani et al., 2009*; *Rochefort et al., 2009*). In addition, we report that this process follows a specific spatial pattern, with the activity of nearby neurons being the most affected.

## Optogenetic IN manipulation confirms developmental increase of inhibition

To substantiate the experimental evidence supporting the developmental increase of inhibition in the mouse mPFC, we optogenetically manipulated IN activity at different stages of early development. To this aim, we selectively transfected Dlx5/6cre and Gad2cre INs with either an excitatory (ChR2, n=19 mice) or an inhibitory opsin (ArchT, n=40 mice) using a combination of mouse lines and viral approaches. Briefly, expression of an excitatory opsin in INs was achieved by injecting P0-1 Dlx5/6cre

and Gad2$^{cre}$ mice with a virus encoding for ChR2 (AAV9-Ef1alpha-DIO-hChR2(ET/TC)-eYFP). Expression of an inhibitory opsin was instead achieved by crossing Dlx5/6$^{cre}$ and Gad2$^{cre}$ mice with a mouse line (Ai40(RCL-ArchT/EGFP)-D) expressing ArchT under a cre-dependent promoter. No significant differences between experiments targeting Dlx5/6$^{cre}$ and Gad2$^{cre}$ neurons were detected and therefore, the datasets were pooled. In line with previously developed protocols (*Bitzenhofer et al., 2020*; *Chini et al., 2020*; *Bitzenhofer et al., 2017b*), we applied a 3-s-long 'ramp-like' optogenetic stimulation of increasing intensity (*Figure 4A*, *Figure 4—figure supplement 1A*).

IN activation led to conspicuous modulation of SUA across all investigated ages. Upon stimulus, a small number of neurons (putative INs) gradually increased their firing rate (*Figure 4B*). The proportion of stimulated INs was similar among mouse lines (main mouse line effect, p=0.14, generalized linear mixed-effect model) and across ages (main age effect = 0.14, 95% CI [–0.04; 0.34], p=0.12, generalized linear mixed-effect model) (*Figure 4—figure supplement 1B*; individual dots correspond to optogenetic stimulation protocols, see figure legend and Materials and methods). This is in line with the histological quantification of the number of virally transfected neurons that led to similar results for all mouse lines (main mouse line effect, p=0.45, linear mixed-effect model) and developmental stages (main age effect = −1.35, 95% CI [−3.22; 0.51], p=0.18, linear mixed-effect model) (*Figure 4—figure supplement 1C*). While putative INs increased their firing rate in response to optogenetic stimulation, a larger proportion of neurons (putative PYRs) significantly decreased their firing rate (*Figure 4B*). In line with the results above that indicated increasing inhibition throughout development, the proportion of inhibited neurons augmented with age (main age effect = 0.31, 95% CI [0.09; 0.55], p=0.005, generalized linear mixed-effect model) (*Figure 4C*; individual dots correspond to optogenetic stimulation protocols, see figure legend and Materials and methods). These results are unlikely to be biased by a 'floor effect' due to the low firing rate of neurons in the youngest mice, as limiting the analysis to neurons in the top 50% for spikes fired during the optogenetic protocol yielded an even stronger effect (main age effect = 0.47, 95% CI [0.17; 0.81], p=0.002, generalized linear mixed-effect model) (*Figure 4—figure supplement 1D*; individual dots correspond to optogenetic stimulation protocols, see figure legend and Materials and methods). Regardless of age, after terminating the optogenetic stimulus, PYRs responded with a prominent 'rebound' increase in firing rate, similar to the effects reported for the adult brain (*Roux et al., 2014*; *Sessolo et al., 2015*). Such widespread inhibition upon IN stimulation supports the hypothesis that GABA exerts an inhibitory population-level effect already during the first postnatal days.

To dissect the main 'neuronal trajectories' in response to light stimulation, we performed principal component analysis (PCA) on trial-averaged smoothed and normalized spike trains and projected the time-varying activity of neurons onto a low-dimensional space. The two principal components captured two populations of neurons that, during optogenetic stimulation, responded with a monotonic decrease (first component) and increase (second component) in firing rate (*Figure 4D*). While opto-tagging at such an early age is not possible (*Weir et al., 2014*), we hypothesize that the first component corresponds to inhibited PYRs and the second one to INs that were activated with the optogenetic stimulus. Interestingly, the two dynamics were strikingly similar across age groups (*Figure 4D*).

To corroborate this hypothesis, we carried out a series of simulations in a LIF neural network model analogous to the one previously described. To mimic the 'ramp-like' optogenetic stimulation, we injected INs with repeated sweeps of an excitatory current with the same temporal profile (3 s of stimulation of increasing intensity) (*Figure 4—figure supplement 1E*). The model confirmed that, similarly to the neuronal trajectory described by the first PCA component, PYRs decreased their firing rate in response to the stimulation, while INs responded with an increase in firing rate that is analogous to the neuronal trajectory described by the second PCA component (*Figure 4—figure supplement 1F*). To test the role played by inhibition in generating the population dynamics, we systematically varied the reversal potential ($V_{rev}$) driving the inhibitory post-synaptic currents. $V_{rev}$s that were higher than the action potential threshold ($V_{thr}$) (excitatory GABA) between the resting membrane potential ($V_{rest}$) and the $V_{thr}$ (depolarizing GABA) or equal to $V_{rest}$ resulted in runaway excitation and average firing rates within 100–1000 Hz range (*Figure 4—figure supplement 1G*). $V_{rev}$ that were at least 5 mV lower than the $V_{rest}$ resulted in stable networks, but only $V_{rev}$s that were 10 mV or lower than $V_{rest}$ were able to fully recapitulate the experimentally observed population trajectories (*Figure 4—figure supplement 1H*).

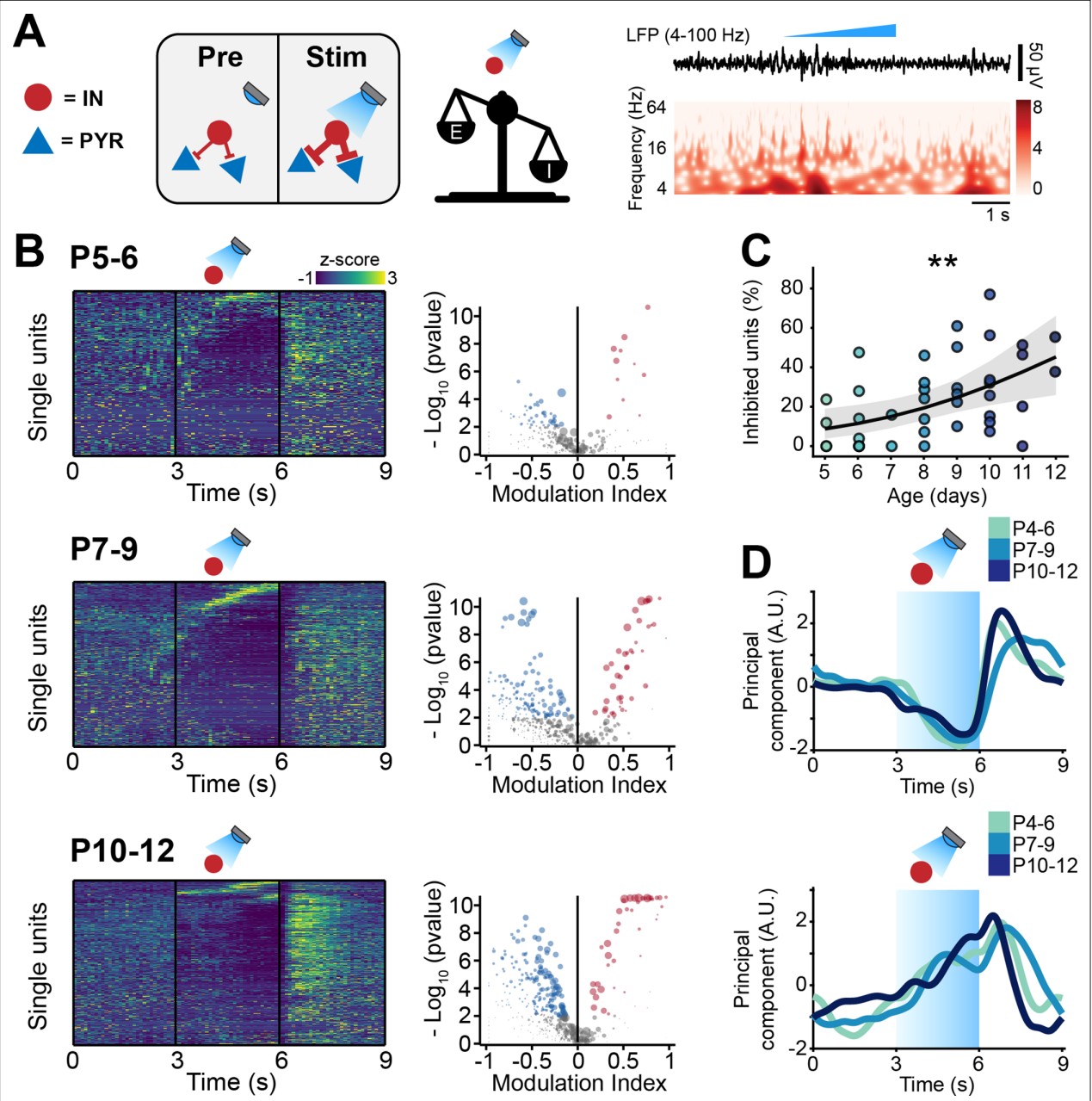

**Figure 4.** Optogenetic stimulation of interneuron (IN) activity leads to widespread inhibition in the developing mouse medial prefrontal cortex (mPFC). (**A**) Schematic representation of the effects induced by optogenetic IN stimulation (left). Representative local field potential (LFP) trace (4–100 Hz band-pass filter) with a corresponding wavelet spectrum at an identical timescale during ramp light stimulation (473 nm, 3 s) of INs in the mPFC of a P10 mouse. (**B**) Z-scored single unit firing rates in response to optogenetic stimulation of INs (left) and volcano plot displaying the modulation index of pre vs. stim single unit firing rates (right) for P4-6 (top, $n$=268 units and 5 mice), P7-9 (middle, $n$=480 units and 7 mice), and P10-12 (bottom, $n$=475 units and 7 mice) mice. Color codes for firing rate. (**C**) Scatter plot displaying the percentage of inhibited units with respect to age ($n$=19 mice). (**D**) First (top, putative pyramidal neurons [PYRs]) and second (bottom, putative INs) principal component analysis (PCA) component of trial-averaged spike trains in response to optogenetic stimulation of INs. Color codes for age group. In (**C**) the regression is presented as mean and 95% CI. Asterisks in (**C**) indicate significant effect of age. **$p<0.01$. Individual dots in (**C**) indicate distinct optogenetic protocols (up to two per mouse, for details see Materials and methods). Generalized linear mixed-effect model (**C**). For detailed statistical results, see *Supplementary file 1*.

The online version of this article includes the following figure supplement(s) for figure 4:

**Figure supplement 1.** Optogenetic manipulation of interneuron (IN) activity affects local field potential (LFP) activity in the mouse developing medial prefrontal cortex (mPFC).

**Figure supplement 2.** Optogenetic manipulation of interneuron (IN) activity in cre- mice does not affect the developing mouse medial prefrontal cortex (mPFC).

Taken together, these data indicate that, while the inhibition strength exerted by INs increases throughout development, the dynamics with which the PYR-IN network responds to IN activation does not change across the first 2 postnatal weeks. These data also provides support to the notion that, on a network level, GABA exerts an inhibitory effect already during the first postnatal week (*Kirmse et al., 2015*; *Murata and Colonnese, 2020*).

To further corroborate these results, we carried out analogous experiments in mice expressing an inhibitory opsin in INs (*Figure 5A*, *Figure 4—figure supplement 1A*).

Light-induced IN silencing resulted in a widespread progressive increase of firing, with very few inhibited neurons (12/1611 neurons) (*Figure 5B*). Independent of age group, the increased firing abruptly returned to baseline levels upon terminating the optogenetic stimulus (*Figure 5B*). The proportion of neurons responding with a firing rate increase during IN inhibition augmented with age (main age effect = 0.54, 95% CI [0.37; 0.74], p<10$^{-8}$, generalized linear model) (*Figure 5C*; individual dots correspond to optogenetic stimulation protocols, see figure legend and Materials and methods). To qualitatively compare the 'neuronal trajectories', we performed trial-averaged PCA. The first component, that captured the activity of putative PYRs, responded to the light stimulus with a monotonic rise of firing rate that quickly dropped as soon as the stimulus stopped (*Figure 5D*). On the contrary, and in line with previous experimental (*Moore et al., 2018*; *Tsodyks et al., 1997*; *Sadeh and Clopath, 2020*) and theoretical work (*Kato et al., 2017*; *Sanzeni et al., 2020*) in the adult brain, putative INs (the trajectory captured by the second component), had a biphasic response. They were initially inhibited but, halfway through the optogenetic stimulation, displayed a steep increase in firing rate, that persisted until end of the stimulus (*Figure 5D*). This response pattern is typical for inhibition-stabilized networks (ISNs), a regime at which adult cortical circuits operate (*Sanzeni et al., 2020*; *Sadeh et al., 2017*), and that has been suggested to not characterize the early developing brain (*Rahmati et al., 2017*).

To corroborate the experimental findings of putative PYRs' and INs' temporal dynamics, we carried out a series of simulations in a LIF neural network model. To mimic the 'ramp-like' optogenetic inhibition of INs, we injected this population of neurons with repeated sweeps of an inhibitory current with the same temporal profile (3 s of stimulation of increasing intensity) (*Figure 5E*). The model confirmed that, similarly to the neuronal trajectory described by the first PCA component, PYRs monotonically increased their firing rate in response to the IN inhibition (*Figure 5F*). INs had a biphasic response analogous to the one described by the second PCA component. They initially decreased their firing rate and then, halfway through the ramp stimulation, robustly increased their firing rate (*Figure 5F*). To test the role of inhibition in generating the population dynamics, we systematically varied the $V_{rev}$ driving the inhibitory post-synaptic currents. Analogously to what has been previously shown, excitatory and depolarizing GABA both resulted in runaway excitation and average firing rates that were in the 100–1000 Hz range (*Figure 4—figure supplement 1I*). $V_{rev}$ that were at least 5 mV lower than $V_{rest}$ resulted in stable networks, but only $V_{rev}$ that were 10 mV or lower than $V_{rest}$ were able to fully recapitulate the population trajectories that we experimentally observed (*Figure 4—figure supplement 1J*).

Thus, while the network excitation derived from IN inhibition increased throughout development, the dynamics with which the PYR-IN network responds to IN inhibition did not change during the first 2 postnatal weeks. These data further support the conclusion that, on a network level, GABA exerts an inhibitory effect already in the first postnatal week (*Kirmse et al., 2015*; *Murata and Colonnese, 2020*).

We have previously shown that the optogenetic paradigm that we utilized does not lead to significant tissue heating (*Bitzenhofer et al., 2017b*), but to further rule out possible nonspecific effects, we applied the same stimulation paradigm to cre- mice (*n*=10 mice, 380 neurons, *Figure 4—figure supplement 2A*). Pooling together all investigated mice, only 6 out of 380 units were activated, whereas none was inhibited. These results are in line with the statistical threshold (0.01) that was used for this analysis (proportion of modulated units 0.016, CI [0.007; 0.034], *Figure 4—figure supplement 2B*). Thus, the used light stimulation leads to minimal non-significant, if any, unspecific modulation of neuronal firing.

Taken together, these data show that optogenetic manipulation of INs robustly affects the neonatal prefrontal network in an age-dependent manner. Stimulating INs induced widespread inhibition of putative PYRs, whereas the contrary was true after IN inhibition. Both effects augmented with age. However, the ability of INs to control the cortical inhibition did not qualitatively change during the first

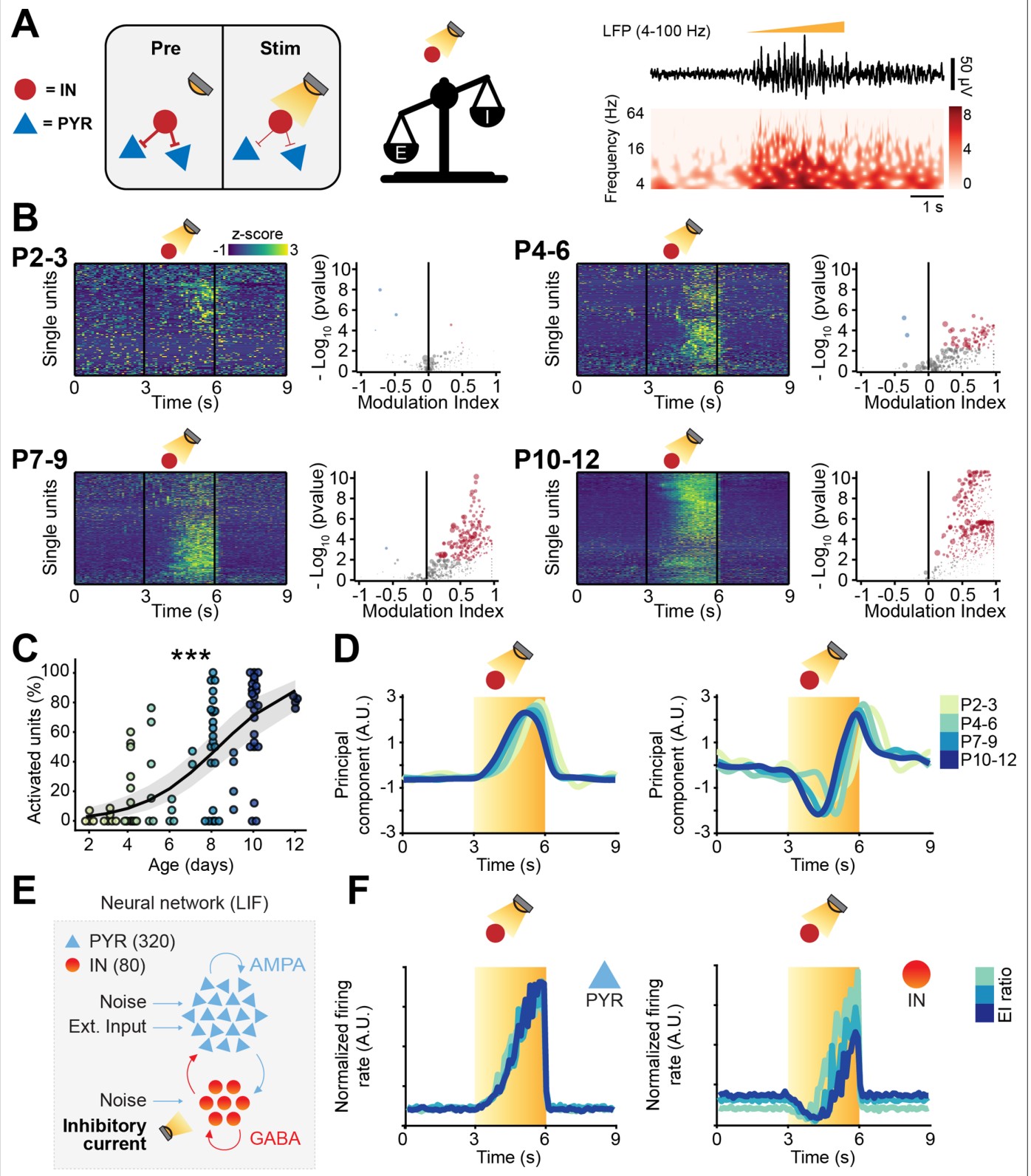

**Figure 5.** Optogenetic inhibition of interneuron (IN) activity leads to widespread excitation in the developing mouse medial prefrontal cortex (mPFC). (**A**) Schematic representation of the effects induced by optogenetic IN inhibition (left). Representative local field potential (LFP) trace (4–100 Hz band-pass filter) with a corresponding wavelet spectrum at an identical timescale during ramp light stimulation (594 nm, 3 s) of INs in the mPFC of a P10 mouse. (**B**) Z-scored single unit firing rates in response to optogenetic stimulation of INs (left) and volcano plot displaying the modulation index of

*Figure 5 continued on next page*

*Figure 5 continued*

pre vs. stim single unit firing rates (right) for P2-3 (top left, *n*=164 units and 5 mice), P4-6 (top right, *n*=286 units and 11 mice), P7-9 (bottom left, *n*=470 units and 13 mice), and P10-12 (bottom right, *n*=691 units and 11 mice) mice. Color codes for firing rate. (**C**) Scatter plot displaying the percentage of activated units with respect to age (*n*=40 mice). (**D**) first (top) and second (bottom) principal component analysis (PCA) component of trial-averaged spike trains in response to optogenetic inhibition of INs. Color codes for age. (**E**) Schematic representation of the neural network model. (**F**) Modeled pyramidal neurons (PYRs) (left) and INs (right) trial-averaged normalized firing rate in response to optogenetic inhibition of INs. Color codes for excitation-inhibition (E-I) ratio. In (**C**) the regression is presented as mean and 95% CI. Asterisks in (**C**) indicate significant effect of age. \*\*\*p<0.001. Individual dots in (**C**) indicate distinct optogenetic protocols (up to two per mouse, see Materials and methods). Generalized linear mixed-effect model (**C**). For detailed statistical results, see ***Supplementary file 1***.

2 postnatal weeks, resembling adult patterns. These data provide evidence against the long-standing hypothesis of network-level excitatory effects of GABA in the developing mouse cortex.

## Optogenetic manipulation of IN activity impacts pairwise spike train correlations

To investigate the relationship between age-dependent dynamics of inhibition and decorrelation of spike trains, we compared STTC before IN optogenetic manipulation (STTCpre) to STTC during optogenetic manipulation (STTCstim). Considering that STTCpre and STTCstim could only be computed in 3 s epochs (times the number of trials), we first verified whether STTCpre was a good predictor of 'baseline' STTC. Pooling across mice and different IN manipulations, STTCpre robustly correlated with baseline STTC across every investigated lag, from 2.5 ms to 1 s (0.66, [0.48; 0.72] median and min-max Pearson correlation; 0.68 [0.40; 0.71] median and min-max Spearman correlation) (***Figure 6—figure supplement 1A-B***). Further, STTCstim exhibited lower correlation values with baseline STTC across all lags, a first hint that optogenetic IN manipulation affected STTC (***Figure 6—figure supplement 1A-B***).

As predicted by the experimental and modeling results, optogenetic modulation of IN activity affected the STTC values across all investigated timescales (***Figure 6A–B***, ***Figure 6—figure supplement 1C***). IN stimulation resulted in decreased STTC values (main IN stimulation effect, $p<10^{-71}$, 1 s lag, linear mixed-effect model) (***Figure 6A***). On the other hand, IN inhibition increased STTC (main IN inhibition effect, $p<10^{-286}$, 1 s lag, linear mixed-effect model) (***Figure 6B***). Moreover, in line with the strongest decorrelation along development for nearby neurons (***Figure 3D***), IN modulation had a larger impact on STTC values of nearby neurons when compared to pairs that are further apart (IN stimulation\*distance interaction, $p=2*10^{-4}$; IN inhibition\*distance interaction $p<10^{-5}$, 1 s lag, linear mixed-effect model) (***Figure 6C–D***).

Thus, these data indicate that IN manipulation causally impacts pairwise correlations between spike trains. The effect of IN manipulation increases with age, in agreement with the notion that inhibition strengthens throughout development.

## Mice with altered developmental E-I ratio have excessively decorrelated activity

Developmental imbalances in E-I ratio have been linked to the pathophysiology of neurodevelopmental disorders (***Antoine et al., 2019***; ***Bitzenhofer et al., 2021***). A corollary of the results above is that impaired developmental E-I ratio should result in altered correlation levels of neuronal activity. To test this hypothesis, we interrogated an open-source dataset that we recently published (***Chini et al., 2020***; ***Bitzenhofer et al., 2021***). The dataset was obtained from extracellular recordings of SUA from the mPFC of P4-10 control and dual-hit genetic-environmental (GE) mice. GE mice mimic the etiology (combined disruption of *Disc1* gene and maternal immune activation) and cognitive impairment of schizophrenia, showing already at neonatal age reduced excitatory activity in the superficial layers of the mPFC (***Chini et al., 2020***; ***Xu et al., 2019***; ***Figure 7A***). On the flipside, deep layers of the mPFC are not affected. Therefore, we hypothesized that GE mice have lower STTC values than controls (i.e. mice lacking the abnormal genetic background and influence of environmental stressor). Considering the layer specificity of the deficits identified in the mPFC of GE mice, we reasoned that this effect should be present in spike trains from neurons in the superficial layers. Overall, GE mice had lower spike train correlations when compared to controls (main condition effect, p=0.032, 1 s lag, linear mixed-effect model) (***Figure 7—figure supplement 1A***). In line with the proposed hypothesis,

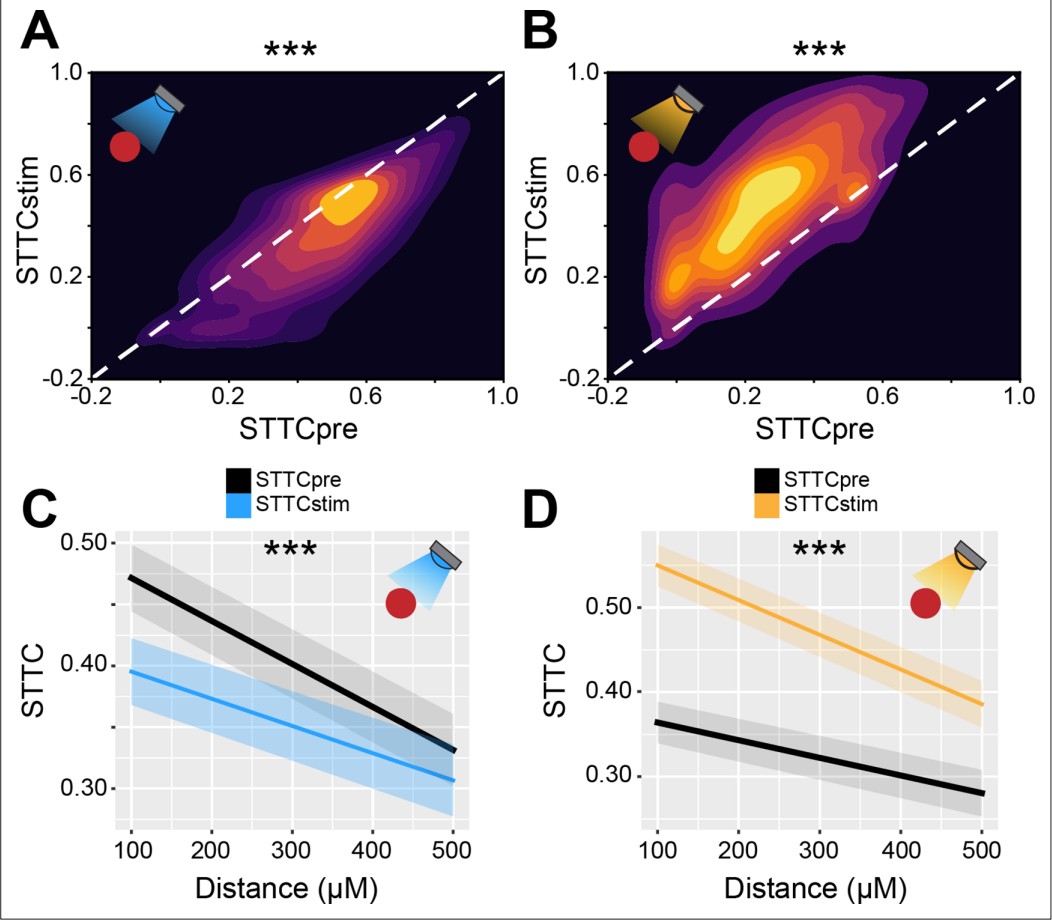

**Figure 6.** Bidirectional optogenetic manipulation of interneuron (IN) activity affects spike time tiling coefficient (STTC) in the developing mouse medial prefrontal cortex (mPFC). (**A and B**) 2D kernel density plots displaying STTC before IN optogenetic manipulation (STTCpre) and STTC during optogenetic manipulation (STTCstim) during IN activation (**A**) and inhibition (**B**) (*n*=10,173 spike train pairs and 19 mice, *n*=9778 spike train pairs and 40 mice, respectively). (**C and D**) Average STTCpre and STTCstim during IN activation (**C**) and inhibition (**D**) over distance (*n*=10,173 spike train pairs and 19 mice, *n*=9778 spike train pairs and 40 mice, respectively). In (**C and D**) data are presented as mean ± SEM. Asterisks in (**A and B**) indicate significant effect of IN activation and inhibition, respectively. Asterisks in (**C and D**) indicate significant effect of IN activation*distance and IN inhibition*distance interaction, respectively. ***p<0.001. Linear mixed-effect models. For detailed statistical results, see ***Supplementary file 1***.

The online version of this article includes the following figure supplement(s) for figure 6:

**Figure supplement 1.** Bidirectional optogenetic manipulation of interneuron (IN) activity effects on spike time tiling coefficient (STTC) depends on spatial configurations and age.

this deficit depended on whether the neuron pair was situated in the superficial or deep layers of the mPFC (condition*layer interaction, p<10⁻⁷, 1 s lag, linear mixed-effect model). While there was no significant difference between STTC of controls and GE spike train pairs situated in the deep layers (p=0.15, 1 s lag, linear mixed-effect model), spike train pairs of GE mice in which one of the two neurons was located in the superficial layers had reduced STTC values (p=0.016, 1 s lag). This difference was even more robust if both neurons were situated in the superficial layers (p=10⁻³, 1 s lag) (***Figure 7B–D***). Last, the effect did not depend on the age of the mouse (condition*age interaction, p=0.16, 1 s lag, linear mixed-effect model) (***Figure 7—figure supplement 1A-B***).

Taken together, these data support the hypothesis that decreased developmental E-I ratio results in reduced spike train pairwise correlations. Further, we show that this effect is remarkably specific. In GE mice, a mouse model characterized by reduced excitatory drive in prefrontal PYRs of the superficial

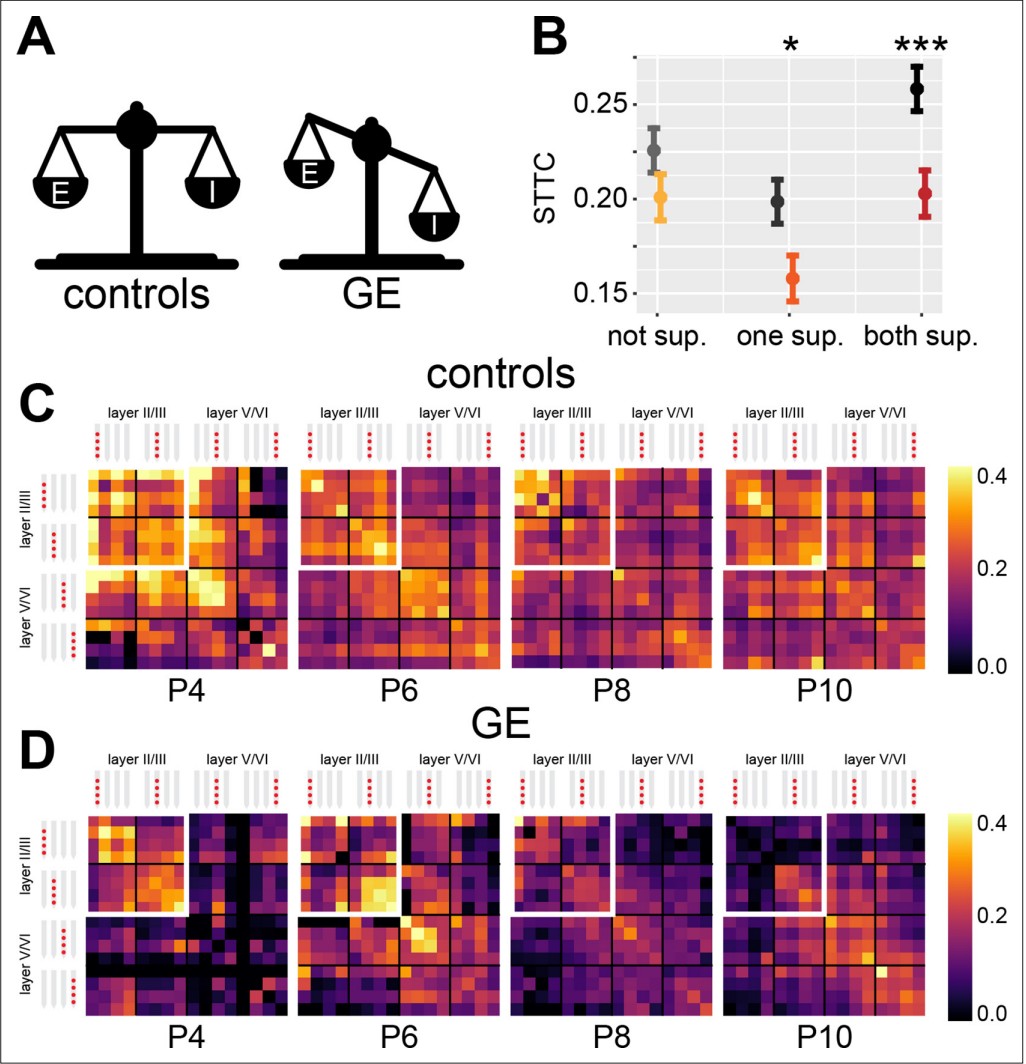

**Figure 7.** Genetic-environmental (GE) mice have reduced spike time tiling coefficient (STTC) values with specific spatial profiles. (**A**) Schematic representation of the excitation-inhibition (E-I) ratio imbalance affecting GE mice. (**B**) STTC of control and GE mice (*n*=18,839 and 11,051 spike train pairs; 33 and 30 mice, respectively) with respect to the number of neurons in the superficial layers in the medial prefrontal cortex (mPFC). (**C**) Weighted adjacency matrices displaying average STTC at 1 s lag of P4, P6, P8, P10, and control mice as a function of the recording sites in which the spike train pair has been recorded (*n*=18,839 spike train pairs and 33 mice). White inset indicates STTC values between spike trains that are located in the superficial layers of the mPFC. Color codes for STTC value. (**D**) Same as (**C**) for GE mice (*n*=11,051 spike train pairs and 30 mice).

The online version of this article includes the following figure supplement(s) for figure 7:

**Figure supplement 1.** Age and spatial profile of spike time tiling coefficient (STTC) values in genetic-environmental (GE) and ES mice.

layers, the reduced correlation levels were largely limited to spike train pairs involving PYRs of the superficial layers.

## E-I ratio decreases with age in newborn babies

Considering the role of E-I ratio for neurodevelopmental disorders, it is of critical relevance to assess whether a developmental strengthening of inhibition occurs also in humans. To this aim, we interrogated two EEG datasets recorded in newborn babies of an age between 35 and 46 post-conception weeks (PCW), a stage of brain development that is roughly equivalent to the one that we studied in mice (*Chini and Hanganu-Opatz, 2021*). While it is not straightforward to compare intracranial

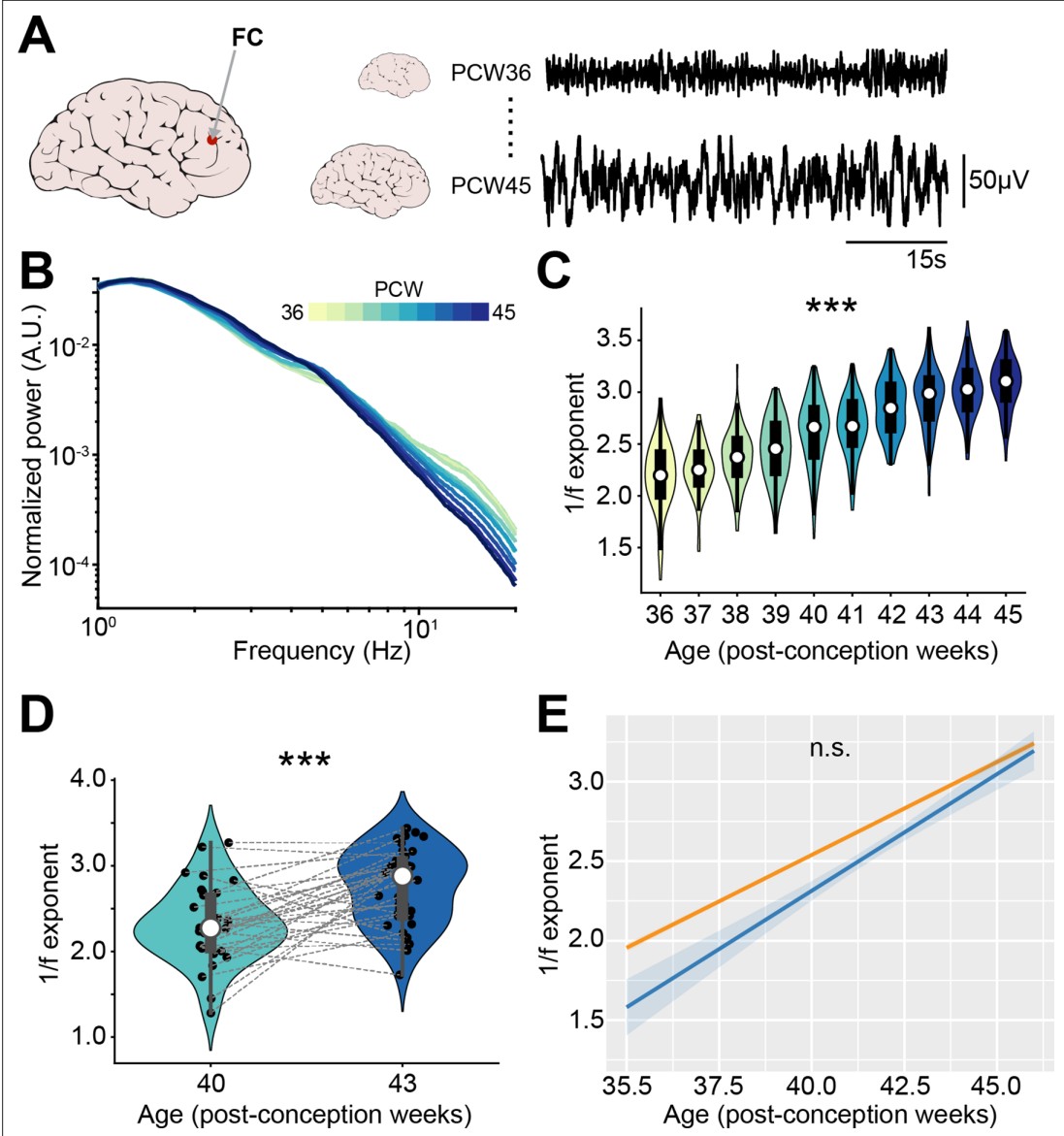

**Figure 8.** 1/f exponent of EEG recordings increases with age in newborn babies. (**A**) Schematic representation of EEG recording from frontal derivations of 36-45 post-conception week (PCW) newborn babies (left) displayed together with representative EEG traces from 36 and 45 PCW newborn babies (right). (**B**) Log-log plot displaying the normalized mean power spectral density (PSD) power in the 1–20 Hz frequency range of 36-45 PCW newborn babies (n=1110 babies). Color codes for age. (**C**) Violin plots displaying the 1/f exponent of 36-45 PCW newborn babies (n=1110 babies). (**D**) Same as (**C**) for 40 and 43 PCW newborn babies (n=72 EEG recordings and 40 babies). (**E**) 1/f exponent over age for the two EEG datasets (n=1110 babies and n=72 EEG recordings and 40 babies, respectively). In (**D**) black dots indicate individual data points. In (**C**) and (**D**) data are presented as median, 25th, 75th percentile, and interquartile range. In (**C**) and (**D**) the shaded area represents the probability distribution density of the variable. In (**B**) and (**E**) data are presented as mean ± SEM. Asterisks in (**C**) and (**D**) indicate significant effect of age. ***p<0.001. Linear model (**C**) and linear mixed-effect models (**D–E**). For detailed statistical results, see *Supplementary file 1*.

The online version of this article includes the following figure supplement(s) for figure 8:

**Figure supplement 1.** EEG power spectral densities (PSDs) of newborn babies.

recordings from a deep structure like the mouse mPFC to human EEG data, to maximize the consistency between approaches, we limited our analysis to channels from the frontal derivations of the EEG (*Figure 8A*).

The first dataset (*Schetinin and Jakaite, 2017*) consisted of 1100 EEG recordings from sleeping babies with an age comprised between 36 and 45 PCW. Similar to the PSDs of recordings from the neonatal mice mPFC, the PSD slope grew steeper over age (*Figure 8—figure supplement 1A*), a

phenomenon that was readily apparent after normalization of the PSD (*Figure 8B*). We quantified the 1/f exponent on the 1–20 Hz frequency range and confirmed that it increased with age (age coefficient = 0.26, 95% CI [0.24; 0.27], p<10⁻¹⁸³, linear model) (*Figure 8C*). A second dataset (*Wielek et al., 2019*) consisted of EEG recordings from 42 sleeping babies, recorded at 40 and 43 PCW. The analyses revealed that also for these data the PSD slope grew steeper (*Figure 8—figure supplement 1B-C*) and the 1/f exponent increased with age (age coefficient = 0.30, 95% CI [0.17; 0.42], p<10⁻⁴, linear mixed-effect model) (*Figure 8D*). The increase in 1/f slope over age was very similar across the two different datasets (mean age coefficients = 0.26 and 0.30) and no statistical difference was found between them (main dataset effect, p=0.15; age*dataset interaction, p=0.21, linear mixed-effect model) (*Figure 8E*).

Thus, the E-I ratio decreases along development also in newborn humans. This feature might represent a fingerprint of cortical circuit maturation in mammalian species.

## Discussion

Integration of INs into the cortical circuitry has been proposed to bear important structural, functional, and behavioral consequences (*Lim et al., 2018*). Here, we show that, even though INs inhibit neuronal activity already in the first postnatal week, the relative strength of the exerted inhibition increases with age, leading to a decrease of E-I ratio. This developmental process contributes to a transition in brain dynamics from early highly synchronous activity patterns to decorrelated neural activity later in life. We further show that an early imbalance in E-I ratio in the mPFC results in altered temporal neural dynamics. Last, leveraging two EEG datasets recorded in newborn babies, we provide evidence for analogous developmental processes taking place in the human cortex.

Inhibitory synaptogenesis is an exquisitely specific process that is orchestrated by distinct molecular programs (*Favuzzi et al., 2019*) and neural activity (*Kepecs and Fishell, 2014*). Its timeline is protracted and it extends to early adulthood in the mouse somatosensory cortex (*Gour et al., 2021*). PYRs are thought of playing an instructive role in this process, by providing molecular cues guiding IN migration and by regulating their survival in an activity-dependent manner (*Lim et al., 2018*; *Wong et al., 2018*). Inhibitory circuits during the first postnatal week have several peculiarities, including a predominance of inhibitory synapses by SST⁺ INs (*Tuncdemir et al., 2016*; *Marques-Smith et al., 2016*; *Guan et al., 2018*) and the presence of hub neurons that control the formation of functional assemblies (*Cossart, 2014*; *Picardo et al., 2011*). One of the unique traits of early inhibitory circuits that has garnered much attention is the hypothesis that GABA might act as an excitatory neurotransmitter. This excitatory action of GABA has been most intensively investigated at single-cell level (*Ben-Ari, 2002*; *Garaschuk et al., 2000*) and has been proposed to result from the low expression of KCC2, a potassium chloride cotransporter extruding chloride (*Ben-Ari, 2002*), which leads to high intracellular chloride concentration in immature neurons. The 'GABA-switch' from an excitatory to inhibitory neurotransmitter has been suggested to take place between the first and the second postnatal week (*Ben-Ari, 2002*). Refuting this hypothesis, recent studies have shown that, already during the first postnatal week, GABAergic transmission, while depolarizing on single neurons, exerts an inhibitory action on population activity (*Che et al., 2018*; *Kirmse et al., 2015*; *Murata and Colonnese, 2020*). In line with this latter interpretation, the present study provides SUA-level evidence of the inhibitory effect of GABA in vivo already in the first postnatal days. This is of particular interest considering that the PFC is a brain area whose development is thought of being more protracted than sensory areas (*Chini and Hanganu-Opatz, 2021*), where the early effects of GABA have been more intensively studied (*Che et al., 2018*; *Kirmse et al., 2015*; *Murata and Colonnese, 2020*). We show that optogenetic inhibition of cortical INs results in a widespread increase of SUA firing rates and, conversely, increasing their activity leads to a reduction of neuronal activity. While the strength of the inhibition exerted by INs increases throughout development, the network effects generated by INs do not qualitatively change with age. Already during the first postnatal week, optogenetic inhibition of INs leads to a paradoxical increase in their firing rate. This dynamics is characteristic of ISNs that have a high degree of recurrent inhibition (*Sadeh and Clopath, 2020*), a regime that has been proposed to emerge at later developmental phases (*Rahmati et al., 2017*). Here, we find experimental and computational evidence that the mPFC might operate in an ISN regime already shortly after birth. Neural network modeling suggests that this is only possible in the presence of a strong hyperpolarizing chloride drive at inhibitory synapses. Of note, we only observed a paradoxical effect, the signature of ISN networks,

in response to optogenetic inhibition and not stimulation of INs. This apparent discrepancy is most likely the result of the approach that we employed to express an inhibitory opsin in INs. The combination of two mouse lines (see Materials and methods) targeted a larger proportion of neurons than the approach that combined a mouse line with viral injections (see Materials and methods) and has been used to express the inhibitory opsin. The proportion of activated/inhibited neurons is a critical factor in determining whether a paradoxical response is elicited (*Sadeh and Clopath, 2020*; *Sanzeni et al., 2020*). Similar discrepancies between the two different targeting approaches have been previously reported for the adult cortex (*Sanzeni et al., 2020*). Further, in neural network models, the paradoxical effect is more readily observed in response to IN inhibition than IN stimulation (*Sadeh and Clopath, 2020*). Thus, while the effects of GABA might differ within brain regions (*Murata and Colonnese, 2020*), our data supports the notion that, in the mPFC, GABA has an inhibitory role already at P2 and that even in the first postnatal days, the cortex operates in an ISN regime.

As the brain develops and inhibitory synaptogenesis progresses, the temporal coordination between excitatory and inhibitory transmission tightens (*Dorrn et al., 2010*; *Moore et al., 2018*) with a relative strengthening of inhibition with respect to excitation (i.e. a decrease in E-I ratio) (*Zhang et al., 2011*). While E-I ratio is thought of being a critical feature of healthy neural networks (*Moore et al., 2018*), the functional consequences of the developmental E-I ratio decrease are still poorly understood. Individually, E-I ratio imbalances (*Trakoshis et al., 2020*; *Antoine et al., 2019*; *Sohal and Rubenstein, 2019*; *Gao and Penzes, 2015*; *Medendorp et al., 2021*; *Ferguson and Gao, 2018*) and altered correlational structure of brain activity (*Luongo et al., 2016*; *Hamm et al., 2017*; *Zick et al., 2018*) have both been linked to mental disorders. Studying how these two processes are linked to each other is therefore likely to be insightful for understanding the pathophysiology of these disorders. To address this knowledge gap, we explored the impact of varying E-I ratio in a biologically plausible neural network model. In line with previous results (*Trakoshis et al., 2020*; *Gao et al., 2017*), we show that E-I ratio can be indirectly tracked by measuring the 1/f exponent, a notion that we leveraged to show that there is an E-I ratio decrease in the mouse and human mPFC. We further show that, in a neural network model, a relative increase in inhibition results in decreased pairwise correlations of spike trains. Confirming the modeling results, we report that, in the mouse mPFC, the inferred E-I ratio decrease taking place across the first two postnatal weeks is accompanied by a reduction in pairwise spike trains correlations, as previously described in sensory cortices (*Golshani et al., 2009*; *Rochefort et al., 2009*; *Cutts and Eglen, 2014*; *Siegel et al., 2012*). To further strengthen the link between the two processes, we bidirectionally manipulated the activity of prefrontal INs by light. In line with our hypothesis, optogenetic stimulation of INs (i.e. decreasing E-I ratio) results in reduced correlations among spike trains. Conversely, optogenetic inhibition of INs (i.e. increasing E-I ratio) increases correlations among spike trains. IN inhibition results in increased spike train correlations even though, in the last portion of the optogenetic protocol, IN displays a paradoxical increase in firing rate. This might indicate that even a transient reduction in inhibition strength might be sufficient to increase neural correlations. Both the age-dependent as well as IN manipulation-induced effects on activity correlations do not impact all spike train pairs in a uniform manner. Rather, neuron pairs that are close to each other are more severely affected than those that are farther apart. While an IN subtype-specific dissection of the mechanisms accounting for neural activity decorrelation is beyond the scope of this study, this effect might be explained by the fact that PV$^+$ INs preferentially provide local inhibition (*Tremblay et al., 2016*) and have a particularly protracted integration into the prefrontal cortical circuitry (*Bitzenhofer et al., 2020*). PV$^+$ INs generally provide inhibition at the soma and the axon initial segment (*Tremblay et al., 2016*), a position that is particularly suited to inhibit the spiking output of PYRs and thereby reduce pairwise spiking correlations. The progressive embedment of PV$^+$ INs in the rodent prefrontal circuitry has also been suggested to be responsible for the increase in the average frequency of the LFP oscillations that are generated by layer 2/3 PYRs (*Bitzenhofer et al., 2020*). Future studies might shed light on whether this process is related to the decorrelation of brain activity.

In the adult brain and modern artificial neural networks, most neurons are only sparsely active and correlations between neurons are low (*Olshausen and Field, 2004*; *Cun et al., 1990*; *Frankle and Carbin, 2019*; *Moreno-Bote et al., 2014*). The relationship between correlations, network activity, and behavioral performance is complex and has been reviewed elsewhere (*Kohn et al., 2016*; *Averbeck et al., 2006*). Briefly, correlated activity has been proposed as limiting the amount of information

that can be encoded and as being energy-inefficient (*Olshausen and Field, 2004*; *Moreno-Bote et al., 2014*; *Lennie, 2003*). Conversely, theoretical and experimental work has shown that correlated neural activity can also enhance the robustness of information transmission (*Averbeck et al., 2006*; *Valente et al., 2021*). In sensory cortices, the decorrelated adult activity patterns result from a developmental sparsification process (*Golshani et al., 2009*; *Rochefort et al., 2009*; *Modol et al., 2020*; *Cutts and Eglen, 2014*; *Mizuno et al., 2021*). Along the same lines, neural activity has also been shown to become less global and more local throughout development (*Siegel et al., 2012*; *Wosniack et al., 2021*). Here, we describe that a similar transition also occurs in a higher-order brain area such as the mPFC. Several non-mutually exclusive mechanisms underlying this phenomenon have been proposed, such as a transition in synaptic plasticity rules (*Wosniack et al., 2021*), changes in NMDA receptor composition (*Mizuno et al., 2021*), a decrease in the input resistance of neurons (*Golshani et al., 2009*), or a combination of these (*Rahmati et al., 2017*). Here, we propose that an increase in inhibition also plays a role in the decorrelation of neural activity.

Further, we show that the 1/f exponent derived from LFP recorded from the mouse mPFC increases along the first two postnatal weeks. Similarly, the 1/f exponent derived from the frontal derivations of EEG recorded from human babies increases between the 36th and the 45th PCW (from ~2 weeks preterm birth to ~7 weeks after birth, when considering 38 weeks as the average length of human pregnancy; *Jukic et al., 2013*). This is the opposite of processes taking place in aging (*Voytek et al., 2015*), between childhood and adulthood (*He et al., 2019*), and even from the 1st to the 7th month of life (*Schaworonkow and Voytek, 2021*). The decline in the 1/f exponent (indicative of increased E-I ratio) occurring between childhood and elderliness can be explained by the decline of brain GABA levels (*Hermans et al., 2018*) and cortical inhibition (*Lissemore et al., 2018*). This is however unlikely to explain the discrepancy between the current study and the effect reported for babies of 1–7 months of age (*Schaworonkow and Voytek, 2021*), an age range that borders the one that we investigated. In age-matched mice, a wave of interneuronal cell death takes place around this age (*Wong et al., 2018*) and might induce an early shift from E-I ratio decrease to E-I ratio increase. Whether a similar process occurs in humans too and might explain the discrepancy between the two studies is still unknown and, due to the ethical and technical limitations of invasive recordings in humans, difficult to address. Directly investigating pairwise spike train correlations is not feasible without resorting to invasive intracranial recordings. It is however noteworthy that, albeit at a different spatial and temporal resolution, the density of phase and amplitude EEG spatial correlations decreases between the last trimester of pregnancy and the first weeks of life (*Omidvarnia et al., 2014*; *Tokariev et al., 2016*). Importantly, an impaired structure of frontal correlations at birth, such as the one induced by birth prematurity, is predictive of impaired neurological performance (*Tokariev et al., 2019*). Future work should address whether changes in E-I ratio also underlie the developmental maturation of these high-level EEG spatial correlations.

Several studies have reported E-I imbalances in the mPFC of mouse models of mental disorders and in patients affected by these diseases (*Trakoshis et al., 2020*; *Sohal and Rubenstein, 2019*; *Medendorp et al., 2021*; *Ferguson and Gao, 2018*; *Hamm et al., 2017*; *Zick et al., 2018*). Following this stream of evidence, we investigated a dataset previously obtained from in vivo electrophysiological recordings from the mPFC of a mouse mimicking the etiology of schizophrenia (*Chini et al., 2020*; *Bitzenhofer et al., 2021*), generated by combining two mild stressors, a genetic and an environmental one (*Chini et al., 2020*; *Xu et al., 2019*). Their synergistic combination results in severe deficits affecting PYRs that reside in the superficial layers of the mPFC. These neurons display a simplified dendritic arborization, a severe reduction in spine density and firing rate (*Chini et al., 2020*; *Xu et al., 2019*). Thus, GE mice have a reduced E-I ratio in the mPFC superficial layers, which leads to diminished STTC values among spike trains. While in sensory areas correlation might limit information carrying capacity, in associative brain areas, like the mPFC, correlations are thought of improving signal readout, and increased correlations have been linked to improved behavioral performance (*Valente et al., 2021*). This data support the hypothesis that imbalances in E-I ratio might be a possible unifying framework for understanding the circuit dysfunction characterizing neuropsychiatric disorders (*Sohal and Rubenstein, 2019*; *Yizhar et al., 2011*; *Nelson and Valakh, 2015*). In this perspective, it is relevant that the development of E-I ratio can also be quantified, albeit indirectly, also from EEG recordings of newborn babies. The fact that two different EEG datasets yield similar estimations for

the age-dependent changes in the 1/f exponent supports the notion that this parameter might be a robust biomarker of E-I ratio development with potential translational relevance.

# Materials and methods
## Data and code availability
LFP and SUA data that were newly generated for this study are available at the following open-access repository: https://gin.g-node.org/mchini/development_EI_decorrelation.

Code supporting the findings of this study is available at the following open-access repository: https://github.com/mchini/Chini_et_al_EI_decorrelation; *Chini, 2021*.

## Experimental models and subject details
All experiments were performed in compliance with the German laws and following the European Community guidelines regarding the research animals use. All experiments were approved by the local ethical committee (G132/12, G17/015, N18/015). Experiments were carried out on C57BL/6J, Dlx5/6-Cre (Tg(dlx5a-cre)1Mekk/J, Jackson Laboratory), Gad2-IRES-Cre (Gad2tm2(cre)Zjh, Jackson Laboratory), and ArchT (Ai40(RCL-ArchT/EGFP)-D, Jackson Laboratory) mice of both sexes. Mice were housed in individual cages on a 12 hr light/12 hr dark cycle, and were given access to water and food ad libitum. The day of birth was considered P0. To inhibit IN activity, mice from the Dlx5/6-Cre and Gad2-IRES-Cre driver lines were crossed with mice from the ArchT reporter line. To stimulate IN activity, P0-P1 mice from the Dlx5/6-Cre and Gad2-IRES-Cre driver lines were injected in the mPFC with a virus encoding for ChR2 (AAV9-Ef1alpha-DIO-hChR2(ET/TC)-eYFP) as previously described (*Xu et al., 2021*). Details on the data acquisition and experimental setup of open-access datasets that were used in this project have been previously published (*Chini et al., 2020*; *Bitzenhofer et al., 2021*; *Schetinin and Jakaite, 2017*; *Wielek et al., 2019*).

## In vivo electrophysiology and optogenetics
### Surgery
In vivo extracellular recordings were performed from the prelimbic subdivision of the mPFC of non-anesthetized P2-12 mice. Before starting with the surgical procedure, a local anesthetic was applied on the mice neck muscles (0.5% bupivacain/1% lidocaine). The procedure was carried out under isoflurane anesthesia (induction: 5%; maintenance: 1–3%, lower for older pups, higher for younger pups). Neck muscles were cut to reduce muscle artifacts. A craniotomy over the mPFC (0.5 mm anterior to bregma, 0.1–0.5 mm lateral to the midline) was performed by first carefully thinning the skull and then removing it with the use of a motorized drill. Mice were head-fixed into a stereotactic frame and kept on a heated (37°C) surface throughout the entire recording. (Opto)Electrodes (four-shank, 4×4 recording sites, 100 μm between recording sites, 125 μm shank distance; NeuroNexus, Ann Arbor, MI) were slowly inserted into the prelimbic cortex, at a depth varying between 1.4 and 2 mm depending on the age of the mouse. A silver wire implanted into the cerebellum was used as ground and external reference. Before signal acquisition, mice were allowed to recover for 30–45 min, to maximize the quality and stability of the recording as well as single units yield.

### Signal acquisition

Extracellular signals were acquired and digitized at a 32 kHz sampling rate after band-pass filtering (0.1–9000 Hz) using an extracellular amplifier (Digital Lynx SX; Neuralynx, Bozeman, MO, Cheetah, Neuralynx, Bozeman, MO).

### Optogenetic stimulation

Optical stimuli were delivered by an Arduino Uno-controlled (Arduino, Italy) diode laser (Omicron, Austria). The delivered light stimuli varied in wavelength (472 or 594 nm) according to the experimental paradigm (IN stimulation and inhibition, respectively). Laser power was titrated before signal acquisition and adjusted to the minimum level that induced reliable spiking response. Typical light power

at the fiber tip was measured in the range of 15–40 mW/mm². Optogenetic stimulations consisted of ramp-like stimuli of 3 s duration as previously described (**Chini et al., 2020**; **Bitzenhofer et al., 2017b**; **Bitzenhofer et al., 2021**; **Bitzenhofer et al., 2017a**). Ramp stimulations were repeated 30–120 times and carried out on the two outmost lateral shanks of the 4-shank electrodes, corresponding to superficial and deep layers of the mPFC. In **Figures 3C and 4C**, **Figure 4—figure supplement 1B and D**, these two distinct optogenetic protocols are plotted separately (statistical analysis is however run on the individual mouse level, see 'Statistical modeling' section).

## Histology

Epifluorescence images of coronal brain sections were acquired postmortem to reconstruct the position of the recording electrode and quantify the amount of eYFP expressing neurons. Only mice in which the electrodes were placed in the correct position were kept for further analysis. eYFP expression was manually quantified on a slide-by-slide basis.

## Neural network modeling

The architecture of the network was set similarly to **Trakoshis et al., 2020**, and is schematically illustrated in **Figure 2A**. The network was composed of a total of 400 conductance-based LIF units, 80% of which were excitatory (E) (N=320) and 20% were inhibitory (I) (N=80). The units in the network were

**Table 1.** Parameters of the leaky integrate-and-fire network.

**Neuron model**

| Parameter | Description | Excitatory cells | Inhibitory cells |
|---|---|---|---|
| $V_L$ | Leak membrane potential | –70 mV | –70 mV |
| $V_{Thr}$ | Spike threshold potential | –52 mV | –52 mV |
| $V_{Res}$ | Reset potential | –59 mV | –59 mV |
| $\tau_{Ref}$ | Refractory period | 2 ms | 1 ms |
| $C_m$ | Membrane capacitance | 500 pF | 500 pF |
| $g_L$ | Membrane leak conductance | 25 nS | 20 nS |
| $\tau_m$ | Membrane time constant | 20 ms | 10 ms |

**Synapse model**

| Parameter | Description | Excitatory cells | Inhibitory cells |
|---|---|---|---|
| $E_{AMPA}$ | Reversal potential (AMPA) | 0 mV | 0 mV |
| $E_{GABA}$ | Reversal potential (GABA) | –80 mV | –80 mV |
| $g_{AMPA}$ | Conductance (AMPA) | lognormal(0, 1)*/50 * nS | lognormal(0, 1)*/50 * nS |
| $g_{GABA}$ | Conductance (GABA) | lognormal(0, 1)*/12 * nS | lognormal(0, 1)*/60 * nS |
| $g_{AMPA,ext}$ | Conductance external input (AMPA) | 0.234 * 5 nS | – |
| $\tau_{AMPA}$ | Time constant of AMPA decay | 2 ms | 1 ms |
| $\tau_{GABA}$ | Time constant of GABA decay | 8 ms | 8 ms |

**Current**

| Parameter | Description | Excitatory cells | Inhibitory cells |
|---|---|---|---|
| Amplitude | Max (final) current amplitude | / | 0.05–0.2 nAmp |
| Duration | Duration of ramp current | / | 3 s |
| Interval | Interval between currents | / | 6 s |
| Sweeps | Number of repetitions | / | 60 |

*The two parameters of the lognormal distribution refer to, respectively, the mean and the standard deviation of the underlying normal distribution.

all connected with each other, with a synaptic weight that was log-normally distributed. Excitatory (E→E, E→I) and inhibitory (I→I and I→E) synapses were mediated by AMPA and GABA conductances, respectively. All baseline parameter values used in the simulations are listed in *Table 1*. All simulations were performed using Brain2 for Python3.7 (*Stimberg et al., 2019*).

The dynamics of each excitatory and inhibitory cell were governed by the following stochastic differential equation:

$$C_m \frac{dV_m}{dt} = -g_L \left( V_m - V_L \right) - g_{AMPA} \left( V_m - E_{AMPA} \right) - g_{GABA} \left( V_m - E_{GABA} \right) + \eta \qquad (1)$$

with

$$\frac{dg_{AMPA}}{dt} = \frac{-g_{AMPA}}{\tau_{AMPA}} \qquad (2)$$

and

$$\frac{dg_{GABA}}{dt} = \frac{-g_{GABA}}{\tau_{GABA}} \qquad (3)$$

where $V_m$ is the membrane potential, $V_L$ is the leak membrane potential, and $E_{AMPA}$ and $E_{GABA}$ denote the AMPA and GABA current reversal potentials, respectively. The synaptic conductance parameters and the corresponding decay time constants are denoted by $g_{AMPA}$, $g_{GABA}$ and $\tau_{AMPA}$, $\tau_{GABA}$, respectively. $\eta$ is a noise term that is generated by an Ornstein-Uhlenbeck process with zero mean. Due to the near-instantaneous rise times of AMPA- and GABA-mediated currents (both typically <0.5 ms), we opted to neglect these in the current simulations. Moreover, synaptic transmission was assumed to be instantaneous (i.e. with zero delay). The excitatory units of the network received an additional external input in the form of AMPA-mediated Poisson spike trains from an external pool of 100 units with a constant spike rate of 1.5 spikes/s.

In order to assess the effect of altered E-I ratio ($g_E/g_I$), we parametrically modulated all excitatory (through multiplication with 25 linearly spaced values from 0.1 to 0.7) and all inhibitory (26 linearly spaced values from 0.2 to 1.2) synaptic conductances. The network was simulated for a duration of 30 s for each of the 25×26 parameter combinations. For each parameter combination, the LFP of the network was computed by taking the sum of the absolute values of the AMPA and GABA currents on all excitatory cells (*Trakoshis et al., 2020*). Neuronal correlation was estimated by means of the STTC (see below), assessed at a lag of 1 s.

To mimic optogenetic modulation of IN activity, we added an external current to the stochastic differential equation regulating the dynamics of inhibitory neurons:

$$C_m \frac{dV_m}{dt} = -g_L \left( V_m - V_L \right) - g_{AMPA} \left( V_m - E_{AMPA} \right) - g_{GABA} \left( V_m - E_{GABA} \right) + I_{Stim} + \eta \qquad (4a)$$

where $I_{stim}$ is a 3-s-long ramp-like inhibitory or excitatory current, administered in repeated sweeps, with an interval of 6 s.

## Electrophysiological analysis

Data were analyzed with custom-written algorithms in the MATLAB and Python environment that are available on the following github repository: https://github.com/mchini/Chini_et_al_EI_decorrelation (copy archived at swh:1:rev:9a07c56f36c80a60a44a6607a5a4061a37d96ae7; *Chini, 2021*).

### Detection of active periods

During early development, brain activity is characterized by an alternation of periods of isoelectric traces (silent periods) and oscillatory bursts (active periods). To detect and quantify the properties of active periods, we developed a novel detection algorithm. For this, the extracellular signal was band-pass filtered (4–20 Hz) and downsampled to 100 Hz, before being averaged across recording electrodes. The average signal (raw and z-scored) was then passed through a boxcar square filter (500 ms) on which a hysteresis threshold was applied. Active periods were firstly detected as oscillatory peaks exceeding an absolute or relative threshold (100 µV or four standard deviations, respectively) and subsequently extended to all neighboring time points that exceed a lower threshold (50 µV or two standard deviations, respectively). The combination of absolute and relative thresholding makes this

approach suitable to a wide range of signals, from the highly discontinuous brain activity of P2 mice, to the nearly continuous brain activity of P11-12 mice (*Figure 1A–B*). Neighboring active periods whose inter-active period interval was shorter than 1 s were merged. Active periods whose duration was smaller than 300 ms were discarded.

### Power spectral density

PSDs for mouse and human data (see below for exception) were computed with the *mtspecgramc* function of the Chronux Toolbox (10-s-long windows, 5 s overlap). Median averaging was the preferred measure of central tendency (*Izhikevich et al., 2018*). To quantify the PSD modulation by IN optogenetic stimulation/inhibition, we computed the MI (see below) of the PSD computed on the last 1.5 s of the optogenetic stimulation with the PSD computed on the 1.5 s preceding stimulus delivery.

### EEG preprocessing

EEG signal was extracted only from frontal electrodes (Fp1, F7, F3, Fp2, F8, F4, Fpz, when available) and re-referenced to a common average reference before further analysis. From the EEG dataset of 1100 sleeping babies (*Schetinin and Jakaite, 2017*), epochs whose average envelope amplitude exceeded two standard deviations from the mean were considered as possible artifacts and were removed from further analysis. No preprocessing was applied to the EEG dataset of sleeping babies recorded at 40 and 43 PCW, as PSDs were already included in the freely available data (*Wielek et al., 2019*).

### 1/f exponent

The 1/f exponent was extracted on the 5–20 and 5–45 Hz (human and mouse data, respectively) frequency range of PSDs using the FOOOF package (*Donoghue et al., 2020*) with the 'fixed' aperiodic mode. To quantify the 1/f exponent modulation by IN optogenetic stimulation/inhibition, we compared the exponent obtained by PSDs computed on the second half of the optogenetic stimulation with the baseline exponent.

### Spike sorting

Spike sorting was performed using Klusta (*Rossant et al., 2016*). Automatically obtained clusters were then manually curated using phy (https://github.com/cortex-lab/phy, *Rossant, 2022*).

### Spike time tiling coefficient

The STTC, a metric that tracks correlations between spike trains and is robust to changes in firing rate, was calculated as previously described (*Cutts and Eglen, 2014*; *Yang et al., 2021*; *Figure 3A*):

$$STTC = \frac{1}{2}\left( \frac{P_A - T_B}{1 - P_A T_B} + \frac{P_B - T_A}{1 - P_B T_A} \right) \tag{4b}$$

where $P_A$ is defined as the proportion of spikes in spike train $A$ that falls within $\pm\Delta t$ of a spike from spike train $B$. $T_A$ is defined as the proportion of time that occurs within (is 'tiled' by) $\pm\Delta t$ from the spikes of spike train $A$. The same applies for $P_B$ and $T_B$. The 'lag' parameter $\pm\Delta t$ controls the 'timescale' at which the STTC is computed, a parameter that we systematically varied across more than three orders of magnitude (from 2.5 ms to 10 s). Baseline STTC analysis was limited to spike trains pairs that were recorded for at least an hour and for which both spike trains had at least 50 spikes (40,921 of 56,613 spike train pairs). To quantify the STTC modulation by IN optogenetic stimulation/inhibition, we compared the STTC derived by spike matrices obtained during the 3 s optogenetic stimulation with the STTC derived by spike matrices obtained during the 3 s preceding optogenetic stimulation.

## Modulation index

The modulation index (MI) is a normalization strategy that was used to compute optogenetically induced changes in firing rate and LFP power. The MI has the desirable property of bounding the normalized value between –1 and 1. MI was computed as:

$$MI = \frac{Value_{PRE} - Value_{STIM}}{Value_{PRE} + Value_{STIM}}$$

## Optogenetic modulation of electrophysiological parameters

Modulation of firing rate by optogenetic manipulation was quantified using the MI and signed-rank testing that compared the firing rate during the last 1.5 s of optical stimulation with the firing rate during the 1.5 s preceding stimulus delivery.

## PCA of spike matrices during optogenetic stimulations

The first two PCA components of the spike during optogenetic stimulations were computed on trial-averaged spike trains that were convolved with a Gaussian window (500 ms length, 50 ms standard deviation) and z-scored across the time dimension.

## Statistical modeling

Statistical modeling was carried out in the R environment. All the scripts and the processed data on which the analysis is based are available from the following github repository: https://github.com/mchini/Chini_et_al_EI_decorrelation (*Chini, 2021*).

Nested data were analyzed with (generalized) linear mixed-effects models (*lmer* and *glmer* functions of the *lme4* R package; *Bates et al., 2014*). Depending on the specific experimental design, we used 'mouse' or 'subject' as random effects. For statistical analysis of STTC, to maximize interpretability of the results (i.e. avoid multiple triple interactions), each STTC lag was investigated as its own independent variable, using identical statistical models. Regression on that data, upon visual inspection seemed to be better fit by an exponential curve, were fitted with generalized linear (mixed-effect) models (family = Gamma, $\alpha$=1, link = inverse). Proportions (e.g. the proportion of activated/inhibited units) were also fitted with generalized linear (mixed-effect) models (family = Binomial, link = logit). Statistical significance for linear mixed-effects models were computed with the *lmerTest* R package (*Kuznetsova et al., 2017*), using the Satterthwaite's degrees of freedom method. When possible, model selection was performed according to experimental design. When this was not possible, models were compared using the *compare_performance* function of the *performance* R package (*Lüdecke et al., 2021b*), and model choice was based on an holistic comparison of AIC, BIC, RMSE, and R2. Model output was plotted with the *plot_model* (type='pred') function of the *sjPlot* R package (*Lüdecke et al., 2021a*); 95% confidence intervals were computed using the *confint* R function. Post hoc analysis with Tukey multiple comparison correction was carried out using the *emmeans* and *emtrends* functions of the *emmeans* R package (*Lenth et al., 2020*).

## Acknowledgements

We thank Amit Marmelshtein, Stefano Panzeri, Giulio Bondanelli, Sebastian Bitzenhofer, Johanna Kostka, Jastyn Pöpplau, and Lingzhen Song for valuable discussions and feedback on the manuscript, and P Putthoff, A Marquardt, and A Dahlmann for excellent technical assistance. This work was funded by grants from the European Research Council (ERC-2015-CoG 681577 to ILH-O), Marie Curie Training Network euSNN (MSCA-ITN-H2020-860563 to ILH-O), Horizon2020 DEEPER 101016787, the German Research Foundation (437610067, 178316478, and 302153259 to ILH-O) and Landesforschungsförderung Hamburg (LFF76, LFF73 to ILH-O).

## Additional information

### Funding

| Funder | Grant reference number | Author |
|---|---|---|
| European Research Council | ERC-2015-CoG 681577 | Ileana Hanganu-Opatz |
| H2020 Marie Skłodowska-Curie Actions | Marie Curie Training Network euSNN MSCA-ITN-H2020-860563 | Ileana Hanganu-Opatz |
| Horizon 2020 Framework Programme | DEEPER 101016787 | Ileana Hanganu-Opatz |
| Deutsche Forschungsgemeinschaft | 437610067 | Ileana Hanganu-Opatz |
| Landesforschungsfoerderung Hamburg | LFF76 | Ileana Hanganu-Opatz |
| Deutsche Forschungsgemeinschaft | 178316478 | Ileana Hanganu-Opatz |
| Deutsche Forschungsgemeinschaft | 302153259 | Ileana Hanganu-Opatz |
| Landesforschungsfoerderung Hamburg | LFF73 | Ileana Hanganu-Opatz |
| American Friends of the Alexander von Humboldt Foundation | Feodor-Lynen Fellowship | Thomas Pfeffer |

The funders had no role in study design, data collection and interpretation, or the decision to submit the work for publication.

### Author contributions

Mattia Chini, Conceptualization, Data curation, Software, Formal analysis, Validation, Investigation, Visualization, Methodology, Writing - original draft, Writing - review and editing; Thomas Pfeffer, Software, Formal analysis, Visualization, Writing - review and editing; Ileana Hanganu-Opatz, Conceptualization, Supervision, Funding acquisition, Project administration, Writing - review and editing

### Author ORCIDs

Mattia Chini (iD) http://orcid.org/0000-0002-5782-9720
Thomas Pfeffer (iD) http://orcid.org/0000-0001-9561-3085
Ileana Hanganu-Opatz (iD) http://orcid.org/0000-0002-4787-1765

### Ethics

Human subjects: No new human data was collected for this study, only open-access datasets were used.

All experiments were performed in compliance with the German laws and following the European Community guidelines regarding the research animals use. All experiments were approved by the local ethical committee (G132/12, G17/015, N18/015).

### Decision letter and Author response

Decision letter https://doi.org/10.7554/eLife.78811.sa1
Author response https://doi.org/10.7554/eLife.78811.sa2

## Additional files

### Supplementary files

- Supplementary file 1. Detailed statistical results.

- MDAR checklist

## Data availability

LFP and SUA data that were newly generated for this study are available at the following open-access repository: https://gin.g-node.org/mchini/development_EI_decorrelation. Code supporting the findings of this study is available at the following open-access repository: https://github.com/mchini/Chini_et_al_EI_decorrelation, (copy archived at swh:1:rev:9a07c56f36c80a60a44a6607a5a4061a37d96ae7).

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
