## [Editor Report]

This manuscript presents a combination of in vivo recording and optogenetic experiments that together with modeling bring findings with important significance: inhibition is functionally present in the newborn frontal cortex having major effects on EEG dynamics. These important findings challenge the view on the switch in GABAergic excitation to inhibition and extend phenomenological observations to human infant EEG data. The strength of evidence is solid, with appropriate methodology used and only minor weaknesses noted regarding the human infant data.

---

## [Decision Letter]

**Decision letter after peer review:**

Thank you for submitting your article "Developmental increase of inhibition drives decorrelation of neural activity" for consideration by *eLife*. Your article has been reviewed by 3 peer reviewers, and the evaluation has been overseen by a Reviewing Editor and Laura Colgin as the Senior Editor. The following individual involved in review of your submission have agreed to reveal their identity: Sampsa Vanhatalo (Reviewer #3).

Essential revisions:

The manuscript by Chini et al., investigates emergent dynamics of neural activity during brain development, focusing on differences in the relative strength of excitatory and inhibitory neurotransmission. Using a combination of in vivo recordings and optogenetic experiments with computer modelling, the authors show that inhibition is functionally present in newborn frontal areas. They also describe how this process is dysfunctional in a mouse model of a neurodevelopmental disorder. The work challenges the simplified view of the switch in GABAergic excitation to inhibition. By phenomenologically comparing rodent and human infant EEG data, the manuscript may provide translational bridges with significant impact for clinical studies.

While there was overall consensus on the value of the study, issues arise in particular with the points listed below. We all agree that you will need to address these points specifically to warrant publication. Please, note that the next review round should reach a consensus.

(1) An issue related with EEG human data and in particular regarding how are they generated and analyzed. First, we are unclear about the real N and the type of recordings obtained from these infants. Second, we need to understand the selection of the frequency band (1-40Hz), since most data suggest that neonates have most of their signal power at <1Hz and there is very little contribution for >20Hz. We feel you should adhere to the most commonly used standards. Finally, we are surprised about the increase in slope with early maturation, which is not in agreement with earlier publications. Importantly, addressing these points is critical for your manuscript to advance to the next step. While we will be open to discuss the option of leaving human data out of the revised version, we feel they represent an important addition that increases the impact.

(2) A second concern is with optogenetic data in Figure 4. We have difficulties in understanding the number of units per mice and per group, as they refer to age. We feel the ranges of ages used should provide a consistent number of samples. We are also unclear about the statistical design, whether it is running longitudinally or not. We would like to see these parts clarified and improved and an appropriate statistical contrast for nested design implemented.

(3) We are unclear about the paradoxical effects of optogenetic activations of INs. This point will require clarification and possibly additional analysis.

(4) Finally, regarding the model we are not completely clear about the assumptions made, in particular regarding lognormal distribution of synaptic connection strengths. We feel that testing the effect of other distributions may improve the conclusion and better support the model.

Please, also go over the specific points raised in the individual reviews and address them all in your revised version and rebuttal letter.

*Reviewer #1 (Recommendations for the authors):*

Overall this is a nice paper which does a good job of exploring an important question using a broad range of different approaches. The focus on PFC and cross-species analysis are particularly novel and important. There are a few points which I feel could be clearer and some issues surrounding data in Figure 4 where a better picture of how the experiments were performed and analysed would be beneficial. Code is made available via GitHub in keeping with open access policy of the journal.

Abstract:

Claim that mechanism behind decorrelation of activity unknown is not strictly accurate, numerous factors have been shown to contribute, including sensory input, synaptic changes and developmental alterations in EGABA

Introduction:

Claim that SOM integrate before PV is not accurate. See, Pangratz-Fuehrer and Hestrin, 2011; Anastasiades et al., 2016; Daw et al., 2007. They make a similar claim in the discussion "Early inhibitory circuits have several peculiarities, including a predominance of inhibitory synapses by SOM+ INs". Again this is not supported by the data. SOM interneurons certainly have an important and unique role in early development. But this is not to say that PV synapses are less numerous or weaker (in fact one of the papers they cite shows that they are much stronger at P12 and have a 20% higher connection probability than SOM cells).

Results:

In the model of the local network do they include changes in GABAergic driving force (i.e reversal potential) or just the conductance? Although they provide evidence that GABA does not appear excitatory during early development, it does not mean that it may not be depolarizing and that EGABA may change across this period. This change could influence their results. Others have shown developmental changes in EGABA within the developing PFC and so this should be taken into account.

Could differences in baseline firing across P4-12 make it harder to detect inhibition at early ages due to a floor effect? Could this contribute in part to their observation?

Were any recordings made at P4 in Figure 4? If not, why not state P5-6 rather than P4-6?

In terms of the mice recorded at different ages. Were mice recorded at all ages in each time window? For example, at P11 there are 2 mice who show a very low modulation index but at P12 all the units seem to be strongly modulated, but there are only 2 data points plotted vs 4 at P11 and 7 at P10. Were only 2 mice recorded at P12, or do the data points overlap at certain ages? Overall, how many mice were recorded at each age?

In the methods for Figure 4 they state that laser power was "adjusted until it gave the desired response" how was this defined? Was there a difference in laser power across the different ages, could this account for differences in inhibition?

Figure S3B How were positive neurons quantified? Is this per slice or per animal?

Discussion:

While the strength of the inhibition exerted by INs increases throughout development, the ability of INs to control cortical inhibition does not qualitatively change with age. Already during the first postnatal week, inhibition of INs leads to a paradoxical increase in their firing rate.

This sentence could be a little clearer.

IN inhibition results in increased spike-train correlations even though, in the last portion of the optogenetic stimulation, IN display a paradoxical increase in firing rate. As could this. Is there a way that you could rephrase? Stimulation typically means to activate cells, whereas you are suppressing them. Even though there is a paradoxical increase in firing this occurs at the network level, this is not due to the direct effect of light and so the term "optogenetic stimulation" is not accurate.

*Reviewer #2 (Recommendations for the authors):*

1. As mentioned in the weakness, the authors should go into more details about the paradoxical effect. Why is not seen for optogenetic activations of INs, only for the optogenetic inactivations? Also, it would be good to bring in some citations of experimental and theoretical work (Sanzeni et al. *eLife*, Sadeh et al. J Neuro 2017).

2. The authors should really put their work in the context of other studies who have measured and analyzed spontaneous activity and discussed how it evolves over time. For e.g. the Lohmann lab proposes the existence of L and H events (low and high participation rate events observed in the primary visual cortex), see Siegel et al. 2012. In a modeling study with the Gjorgjieva lab (Wosniack et al. *eLife* 2021), they proposed a different mechanism that can lead to the desynchronization (or sparsification) of activity during development, where L events increase in frequency while H events increase. This should be at least discussed. In Leighton et al. (Curr Biol 2021) the Lohmann lab also talked about the role of inhibition (from SOMs) in development. Finally, an interesting study that should be discussed is Rahmati et al. (Sci Rep 2017) which also presents results on sparsification of neural activity in development and the connection to inhibition stabilization.

3. Can the authors discuss the use of lognormal weights in the model? What happens if they are constant or taken from a lognormal distribution? I don't doubt they come from a lognormal distribution in the real circuit, but it would be important for the model to point out why this is important, as many other modeling papers ignore this fact.

4. The authors should present a more extensive discussion of why decorrelation is something that the network might strive to archive and how this relates to the onset of sensory experience and the efficient processing of sensory information.

*Reviewer #3 (Recommendations for the authors):*

My review at this phase will only focus on the few items that I hope would help the authors strengthen the work:

(1) In your introduction, you note that E-I ratio is important in your context because it "is the hallmark of neurodevelopmental disorders, such as autism or schizophrenia". Please note that you are mixing periods in the lifespan: your work is on neonatal brain development, while those disorders are about toddler age (autism) or much much later in life (schizophrenia).

(2) The work is very strong with the case on early inhibition. I find it a bit confusing how the work starts from making a case why the slope of PSD curve (1/f exponent) should be taken as a relevant measure of E-I.

Why not move this part towards the end, just before you introduce the human data? After all, this component appears to have value mainly because it allows you to link your findings on inhibition to the human dataset.

(3) I find it a bit perplexing that you show increase in slope with early maturation. This is opposite to what has been published earlier, and what is the general finding among clinicians. The early EEG (from prematurity to the end of neonatal period) is characterized by a rapid/robust decline in the lowest frequency power ->this translates directly to a decrease of slope.

So there is something unexpected here?

(4) I would also like to understand why you select to analyse 1-40Hz while recent papers have clearly indicated that (i) neonates have most of their signal power <1hz, and (ii) there is very little to be found >20hz.

(5) The human dataset is elusive: You tell that you had N=1100 and N=42 infants (Figure 8 N=1110?`). This would be the by far largest newborn dataset ever published. BUT the papers you cite only have 71+42+40=153 EEG recordings (assuming that they are from different infants). Also, there is no information about the kind of recordings done from these infants.

So, in brief, the information about human data is virtually missing; please elaborate.

[Editors' note: further revisions were suggested prior to acceptance, as described below.]

Thank you for resubmitting your work entitled "An increase of inhibition drives the developmental decorrelation of neural activity" for further consideration by *eLife*. Your revised article has been evaluated by Laura Colgin (Senior Editor) and a Reviewing Editor.

The manuscript has been improved but there are some remaining issues that need to be addressed, as outlined below:

– Regarding human data: while we were overall positive, we still feel the human data may require some clarification in view of the previous concerns. We appreciated your argument that there is no change of slope caused by contribution at 20-40Hz, but they were based on mouse data only (Figure 1-FS1). If you could provide some sort of additional/control analysis on human data as supplementary material, we feel that could help. We would like to stress this is just advice that we leave to your consideration. Importantly, we feel that it would be useful to discuss your observations on changes in slope with early maturation in the context of earlier publications. Please, be sure you add text to the discussion addressing these and previously raised issues and caveats regarding human results.

– In terms of the optogenetic data in Figure 4, we feel some additional clarification is required specifically regarding the way data is represented (per trial not per mice), and potential issues of low N.

*Reviewer #1 (Recommendations for the authors):*

The authors seem to have addressed my previous comments on an earlier version of this manuscript. I still think their statements regarding the predominance and importance of SST cells are a little strong, and largely unnecessary given they don't study them in this paper, but I guess it is still a matter of debate within the field.

In terms of the optogenetic data in Figure 4, their explanation makes sense. It does however seem a little strange to plot 2 trials from the same animal as separate data points. Their analysis seems to account for this, but it does mean that the N is a little low for some time points. That said their new analysis accounting for only the top 50% of active units seems to show a very robust effect consistent with their observations and overall model of inhibition's role in the early network.

*Reviewer #2 (Recommendations for the authors):*

The authors have appropriately addressed all of my, and the other reviewers', comments. There are still a few typos which I'm sure will be fixed in the final version (e.g. line 707 in the version with tracked changes "such AS a transition in synaptic plasticity rules", line 653 the word In's should be INs (all capital)). Overall, this is a very nice paper that people in the field will enjoy reading.

[Editors' note: further revisions were suggested prior to acceptance, as described below.]

Thank you for resubmitting your work entitled "An increase of inhibition drives the developmental decorrelation of neural activity" for further consideration by *eLife*. Your revised article has been evaluated by Laura Colgin (Senior Editor) and a Reviewing Editor.

The manuscript has been improved but there are some remaining issues that need to be addressed, as outlined below:

As indicated in the previous decision letter, the editorial consultation on the issue regarding human data agreed to request that this would be addressed directly in the discussion. The decision letter stated "Importantly, we feel that it would be useful to discuss your observations on changes in slope with early maturation in the context of earlier publications. Please, be sure you add text to the discussion addressing these and previously raised issues and caveats regarding human results.". Please, consider this point carefully when providing a revised version. We specifically ask for the issues raised by the non-responding reviewer to be explicitly addressed in the manuscript. Please also note that *eLife* publishes reviews and decision letters together with manuscripts, so we prefer not to leave important issues unaddressed that were previously raised during reviews and consultation.

---

## [Author Response]

Essential revisions:The manuscript by Chini et al., investigates emergent dynamics of neural activity during brain development, focusing on differences in the relative strength of excitatory and inhibitory neurotransmission. Using a combination of in vivo recordings and optogenetic experiments with computer modelling, the authors show that inhibition is functionally present in newborn frontal areas. They also describe how this process is dysfunctional in a mouse model of a neurodevelopmental disorder. The work challenges the simplified view of the switch in GABAergic excitation to inhibition. By phenomenologically comparing rodent and human infant EEG data, the manuscript may provide translational bridges with significant impact for clinical studies.While there was overall consensus on the value of the study, issues arise in particular with the points listed below. We all agree that you will need to address these points specifically to warrant publication. Please, note that the next review round should reach a consensus.(1) An issue related with EEG human data and in particular regarding how are they generated and analyzed. First, we are unclear about the real N and the type of recordings obtained from these infants. Second, we need to understand the selection of the frequency band (1-40Hz), since most data suggest that neonates have most of their signal power at <1Hz and there is very little contribution for >20Hz. We feel you should adhere to the most commonly used standards. Finally, we are surprised about the increase in slope with early maturation, which is not in agreement with earlier publications. Importantly, addressing these points is critical for your manuscript to advance to the next step. While we will be open to discuss the option of leaving human data out of the revised version, we feel they represent an important addition that increases the impact.

We apologize for the confusion created by the wrong reference that references only part of the open access dataset. We corrected the reference.

We clarified that for the EEG analysis, we already only considered the 1-20 Hz band, whereas 1-40 Hz band was used only for LFP data. We agree with the observation that in the immature brain, most of the PSD power resides at ultra-slow frequencies. However, we do not quantify the PSD power, but rather estimate the slope of the PSD. For this measure, it is less relevant, where most PSD power resides. On the contrary, it has been suggested to choose frequency bands that are “uncorrupted by oscillatory peaks” (Gao et al., 2017). Further, as shown in Figure 1D and Figure 1—figure supplement 1F, there is no change in the slope between 20 and 40 Hz. Thus, the 1/f slope estimation is not biased by the specific frequency range that we chose.

(2) A second concern is with optogenetic data in Figure 4. We have difficulties in understanding the number of units per mice and per group, as they refer to age. We feel the ranges of ages used should provide a consistent number of samples. We are also unclear about the statistical design, whether it is running longitudinally or not. We would like to see these parts clarified and improved and an appropriate statistical contrast for nested design implemented.

In the revised version, we added the required information (see reply to reviewers). We have also provided additional analysis (Figure 4—figure supplement 1D) and a figure (Figure 1) for the reviewers that addresses some of the criticism that was raised with respect to the optogenetic experiments / analyses.

To analyze these experiments, we used a generalized linear mixed-effect model (logit link) with mouse as random effect, and estimated the overall effect that age has on the proportion of units that increase/decrease their firing rate in response to optogenetics. Compared to ANOVAs, generalized linear mixed-effect models estimate the regression coefficients while explicitly taking in consideration the nested nature of the data (i.e. multiple single units per mouse), while allowing more freedom with respect to missing datapoints and the distribution of the data (in this case, since it is not normally distributed as it is bounded between 0 and 1, we used a logit link).

(3) We are unclear about the paradoxical effects of optogenetic activations of INs. This point will require clarification and possibly additional analysis.

We have provided further modeling analysis (Figure 4—figure supplement 1G-J) that strengthens the conclusions of the manuscript with respect to this point. Further, we have expanded the discussion of the paradoxical effect.

(4) Finally, regarding the model we are not completely clear about the assumptions made, in particular regarding lognormal distribution of synaptic connection strengths. We feel that testing the effect of other distributions may improve the conclusion and better support the model.

We have clarified our reasoning behind some of the assumptions that went into the model and have provided a figure for the reviewers that highlights how the effects that we report do not depend on the lognormality of the synaptic weights.

Please, also go over the specific points raised in the individual reviews and address them all in your revised version and rebuttal letter.Reviewer #1 (Recommendations for the authors):Overall this is a nice paper which does a good job of exploring an important question using a broad range of different approaches. The focus on PFC and cross-species analysis are particularly novel and important. There are a few points which I feel could be clearer and some issues surrounding data in Figure 4 where a better picture of how the experiments were performed and analysed would be beneficial. Code is made available via GitHub in keeping with open access policy of the journal.

We thank the reviewer for the constructive feedback and most helpful comments and suggestions.

Abstract:Claim that mechanism behind decorrelation of activity unknown is not strictly accurate, numerous factors have been shown to contribute, including sensory input, synaptic changes and developmental alterations in EGABA

We agree with the reviewer that the sentence was not entirely accurate. We have rephrased it to better reflect the fact that, while a lot of what explain the developmental decorrelation of neural activity still remains to be understood, the underlying mechanisms are not entirely unknown (lines 30-31).

Introduction:Claim that SOM integrate before PV is not accurate. See, Pangratz-Fuehrer and Hestrin, 2011; Anastasiades et al., 2016; Daw et al., 2007. They make a similar claim in the discussion "Early inhibitory circuits have several peculiarities, including a predominance of inhibitory synapses by SOM+ INs". Again this is not supported by the data. SOM interneurons certainly have an important and unique role in early development. But this is not to say that PV synapses are less numerous or weaker (in fact one of the papers they cite shows that they are much stronger at P12 and have a 20% higher connection probability than SOM cells).

We rephrased the sentences in Introduction and Discussion (lines 79-81, 605-607) to highlight that the statement on the predominance of SOM^+^ INs holds true for the first postnatal days (the beginning of the developmental phase that we investigated in this study). At this early stage, SOM+ interneurons have much larger synapses and higher connectivity than PV+ INs (Gao et al., 2017). Pangratz-Fuehrer and Hestrin, 2011 reports no (FS) PV-PYR connectivity until P4, and little connectivity also at P5-6. Daw et al., 2007 reports no functional recruitment of (FS) PV+ INs until P6-7.

Results:In the model of the local network do they include changes in GABAergic driving force (i.e reversal potential) or just the conductance? Although they provide evidence that GABA does not appear excitatory during early development, it does not mean that it may not be depolarizing and that EGABA may change across this period. This change could influence their results. Others have shown developmental changes in EGABA within the developing PFC and so this should be taken into account.

We thank the reviewer for the insightful comment that has spurned us to extend our modeling analysis. As depicted in the novel Figure 4—figure supplement 1G-J, we show that in our model both depolarizing (chloride reversal potential between the resting membrane potential and the action potential threshold) as well as excitatory (chloride reversal potential more positive than the action potential threshold) GABA result in runaway excitation. Further, we also show that even a chloride reversal potential that is equal to or only slightly more negative to the resting membrane potential fails to replicate the population trajectories that we empirically observed in the optogenetic experiments. Only a chloride reversal potential that is 10 mV or more negative than the resting membrane potential fully recapitulates the experimental data.

Could differences in baseline firing across P4-12 make it harder to detect inhibition at early ages due to a floor effect? Could this contribute in part to their observation?

In line with the reviewer’s suggestion, we tested whether a floor effect might have partially explained our results. When we limited the analysis to neurons that were in the top 50% for spikes fired during the optogenetic protocol, the proportion of inhibited units as a function of age was higher than when considered the entire dataset. Therefore, a floor effect as cause of the described results seems unlikely. We added the novel analysis to the manuscript (lines 285-288) and displayed the results in Figure 4—figure supplement 1D.

Were any recordings made at P4 in Figure 4? If not, why not state P5-6 rather than P4-6?

The reviewer is correct, no mouse was recorded at P4. We corrected the figure.

In terms of the mice recorded at different ages. Were mice recorded at all ages in each time window? For example, at P11 there are 2 mice who show a very low modulation index but at P12 all the units seem to be strongly modulated, but there are only 2 data points plotted vs 4 at P11 and 7 at P10. Were only 2 mice recorded at P12, or do the data points overlap at certain ages? Overall, how many mice were recorded at each age?

As specified in Materials and methods (lines 759-771), each mouse has been acutely recorded and only once.

Of the 19 mice that were recorded for the Chr2 experiments: 2 mice were recorded at P5, 3 mice were recorded at P6, 1 mouse was recorded at P7, 3 mice were recorded at P8, 3 mice were recorded at P9, 4 mice were recorded at P10, 2 mice were recorded at P11, 1 mouse was recorded at P12. The individual dots (some were indeed overlapping) do not refer to single mice but rather to individual optogenetic stimulations. The optogenetic protocol was applied twice in each mouse, on the two outmost shanks of the 4-shank electrode (corresponding to superficial and deep layers of the mPFC). We added the missing information to the legend of Figure 4, 5, Figure 4—figure supplement 1, and Materials and methods (lines 781-784).

Of note, the statistical modeling (generalized linear mixed-effect model with mouse as nesting factor) explicitly takes the nesting of the data into account and estimates the parameter of the model (and its significance) on an individual mouse level. We limited the statistical analysis to the estimation of the main effect of age, thereby rending unnecessary recording a large number of mice at every individual postnatal day.

In the methods for Figure 4 they state that laser power was "adjusted until it gave the desired response" how was this defined? Was there a difference in laser power across the different ages, could this account for differences in inhibition?

We rephrased the sentence and specified that the “desired response” was reliable light-induced spiking (lines 778-779). Two pieces of evidence let us assume that it unlikely that different laser power across ages accounts for the differences in inhibition: (i) across the two optogenetic conditions, laser power showed a weak and non-significant (main age effect=-0.23, 95% C.I. [-0.50; 0.04], p=0.10; linear mixed-effect model) (Author response image 1), trend towards decreased intensity as a function of age. This weak trend, if anything, should work against the increase in inhibition that we report; (ii) laser power and optogenetic effect (% of neurons with increased firing rate) were entirely uncorrelated (main laser power effect=-0.03, 95% C.I. [-0.18; 0.18], p=0.74 for ChR2 experiments; main laser power effect=0.03, 95% C.I. [-0.24; 0.29], p=0.81 for ArchT experiments; generalized linear mixed-effect model) (Figure for reviewer 1B-C) in both optogenetic conditions.

**Author response image 1. sa2fig1:** Laser power across age and its effect on neuronal activity. (A) Scatter plot displaying laser power with respect to age (n=59 mice). (B) Scatter plot displaying the percentage of activated units in ChR2 experiments with respect to laser power (n=19 mice). (C) Same as (B) for ArchT experiments (n=40 mice).

Figure S3B How were positive neurons quantified? Is this per slice or per animal?

We specified that eYFP-expressing neurons were manually quantified on a slide-by-slide basis (lines 785-788) and edited the y-axis label on figure S3.

Discussion:While the strength of the inhibition exerted by INs increases throughout development, the ability of INs to control cortical inhibition does not qualitatively change with age. Already during the first postnatal week, inhibition of INs leads to a paradoxical increase in their firing rate.This sentence could be a little clearer.

We rephrased the sentence (lines 621-623).

IN inhibition results in increased spike-train correlations even though, in the last portion of the optogenetic stimulation, IN display a paradoxical increase in firing rate. As could this. Is there a way that you could rephrase? Stimulation typically means to activate cells, whereas you are suppressing them. Even though there is a paradoxical increase in firing this occurs at the network level, this is not due to the direct effect of light and so the term "optogenetic stimulation" is not accurate.

We rephrased the sentence (lines 658-660).

Reviewer #2 (Recommendations for the authors):1. As mentioned in the weakness, the authors should go into more details about the paradoxical effect. Why is not seen for optogenetic activations of INs, only for the optogenetic inactivations? Also, it would be good to bring in some citations of experimental and theoretical work (Sanzeni et al. eLife, Sadeh et al. J Neuro 2017).

In line with the reviewer’s suggestion, we extended the available discussion of the paradoxical effect in response to IN inhibition referring to the work by Sanzeni and Sadeh (lines 367-373, 633-637).

2. The authors should really put their work in the context of other studies who have measured and analyzed spontaneous activity and discussed how it evolves over time. For e.g. the Lohmann lab proposes the existence of L and H events (low and high participation rate events observed in the primary visual cortex), see Siegel et al. 2012. In a modeling study with the Gjorgjieva lab (Wosniack et al. eLife 2021), they proposed a different mechanism that can lead to the desynchronization (or sparsification) of activity during development, where L events increase in frequency while H events increase. This should be at least discussed. In Leighton et al. (Curr Biol 2021) the Lohmann lab also talked about the role of inhibition (from SOMs) in development. Finally, an interesting study that should be discussed is Rahmati et al. (Sci Rep 2017) which also presents results on sparsification of neural activity in development and the connection to inhibition stabilization.

Already in the previous version of the manuscript, a substantial part of the Introduction and Discussion has been dedicated to put the new results into the context of previous studies done by Rochefort, Lohmann, Colonnese and Khazipov labs. As suggested, we extended the discussion of the mentioned studies, highlighting the developmental decorrelation of neural activity and other potential underlying mechanisms (lines 673-686).

3. Can the authors discuss the use of lognormal weights in the model? What happens if they are constant or taken from a lognormal distribution? I don't doubt they come from a lognormal distribution in the real circuit, but it would be important for the model to point out why this is important, as many other modeling papers ignore this fact.

Abundant literature data have shown that synaptic weights are log-normally distributed in the adult mammal brain (reviewed in Barbour et al., 2007; Buzsáki and Mizuseki, 2014; Scheler, 2017) and, while evidence is not yet conclusive, this log-normality does not seem to arise from activity-dependent processes (Hazan and Ziv 2020). Thus, it is reasonable to assume that synaptic weights are log-normally distributed also in early development. We therefore chose to use log-normally distributed synaptic weights to use biologically plausible parameter choices. We added this explanatory note to the manuscript (lines 173-174).

Further, to verify that the results do not depend on a specific parameter choice, we performed additional modeling simulations with normally distributed synaptic weights. As shown in Author response image 2, the results were similar to those included in the manuscript.

**Author response image 2. sa2fig2:** Figure 2.Increased inhibition leads to an increase in the 1/f exponent and decorrelates spike trains in a neural network model. (A) Schematic representation of the neural network model. (B) Log-log plot displaying the normalized median PSD power in the 30-100 Hz frequency range for varying level of inhibition. Color codes for inhibition strength. (C) Scatter plot displaying the 1/f exponent as a function of net inhibition strength. (D) Scatter plot displaying average STTC as a function of net inhibition strength. For (C) and (D) color codes for inhibition strength with fixed excitation level.

4. The authors should present a more extensive discussion of why decorrelation is something that the network might strive to archive and how this relates to the onset of sensory experience and the efficient processing of sensory information.

We added a new paragraph to the manuscript (lines 673-686).

Reviewer #3 (Recommendations for the authors):My review at this phase will only focus on the few items that I hope would help the authors strengthen the work:(1) In your introduction, you note that E-I ratio is important in your context because it "is the hallmark of neurodevelopmental disorders, such as autism or schizophrenia". Please note that you are mixing periods in the lifespan: your work is on neonatal brain development, while those disorders are about toddler age (autism) or much much later in life (schizophrenia).

We previously showed that, while the disease-related symptoms emerge during later development, the disturbance of neural circuits is initiated already at neonatal age (Hartung et al., 2016; Chini et al., 2020). As suggested, we rephrased the sentence to better highlight this aspect (lines 71-72).

(2) The work is very strong with the case on early inhibition. I find it a bit confusing how the work starts from making a case why the slope of PSD curve (1/f exponent) should be taken as a relevant measure of E-I.Why not move this part towards the end, just before you introduce the human data? After all, this component appears to have value mainly because it allows you to link your findings on inhibition to the human dataset.

We agree with the reviewer that the optogenetic experiments (Figure 3 and 4) strengthen the hypothesis that E-I ratio shifts towards inhibition during development that was proposed in line with the 1/f results presented in Figure 1. However, taken in isolation, we do not think that they are sufficient to make a strong case for the developmental E-I ratio shift. The results in Figure 3-4 suggest that inhibition increases throughout development. This is not the same as a shift in E-I ratio (relative strengthening of inhibition with respect to excitation).

(3) I find it a bit perplexing that you show increase in slope with early maturation. This is opposite to what has been published earlier, and what is the general finding among clinicians. The early EEG (from prematurity to the end of neonatal period) is characterized by a rapid/robust decline in the lowest frequency power ->this translates directly to a decrease of slope.So there is something unexpected here?

To the best of our knowledge, there is only one other paper (Schaworonkow and Voytek, 2021) that has been published on the development of the 1/f exponent at such early stages. Of note, even this paper investigates a later developmental phase (1 to 7 months) when compared to the time window of 1-month prematurity to 1-month of age investigated in the present study. The Discussion includes several hypotheses as to why the two studies might diverge (lines 694-698).

Further, the fact that the slope becomes steeper over this period does not contradict the observation that the early EEG is characterized by a robust decline at the lowest frequency range. Our data solely indicate that the steepness of this decline increases over the two months that are covered by the two datasets. The entire range (1-month prematurity to 1-month of age) can be categorized as (very) early EEG.

(4) I would also like to understand why you select to analyse 1-40Hz while recent papers have clearly indicated that (i) neonates have most of their signal power <1hz, and (ii) there is very little to be found >20hz.

We agree with the observation that in the immature brain, most of the PSD power resides at ultra-slow frequencies. Therefore, we do not quantify the PSD power, but rather estimate the slope of the PSD. For this measure, it is not important to include frequencies at which most power resides. On the contrary, it has been suggested to choose frequency bands that are “uncorrupted by oscillatory peaks” (Gao et al., 2017). As shown in Figure 1D and Figure 1—figure supplement 1F, there is no change in the slope between 20 and 40 Hz. Thus, the 1/f slope estimation is not biased by the specific frequency range that we chose.

Further, we extended the frequency range up to 40 Hz to compute the 1/f exponent only on mouse data (for newborn babies we the cutoff was 20 Hz, see Materials and methods) and, how it is observable from Figure 1D and Figure 1—figure supplement 1F, there is no discontinuity in the rate with which power decays between 20 and 40 Hz. Thus, the 1/f slope estimation is not affected by the specific frequency range that we chose.

(5) The human dataset is elusive: You tell that you had N=1100 and N=42 infants (Figure 8 N=1110?`). This would be the by far largest newborn dataset ever published. BUT the papers you cite only have 71+42+40=153 EEG recordings (assuming that they are from different infants). Also, there is no information about the kind of recordings done from these infants.So, in brief, the information about human data is virtually missing; please elaborate.

We thank the reviewer for spotting the inconsistency with respect to the reference for the first of the two human datasets that we analyze in the manuscript. We apologize for the confusion created by the wrong reference that references only part of the open access dataset. While the two references that were previously given for the dataset are the two that are available on figshare (where the repository is located), they indeed reference only part of the dataset. We have updated the reference by providing one that pertains to a paper describing the entirety of the dataset (1100 recordings) (Schetinin and Jakaite, 2017).

[Editors' note: further revisions were suggested prior to acceptance, as described below.]

The manuscript has been improved but there are some remaining issues that need to be addressed, as outlined below:– Regarding human data: while we were overall positive, we still feel the human data may require some clarification in view of the previous concerns. We appreciated your argument that there is no change of slope caused by contribution at 20-40Hz, but they were based on mouse data only (Figure 1-FS1). If you could provide some sort of additional/control analysis on human data as supplementary material, we feel that could help. We would like to stress this is just advice that we leave to your consideration. Importantly, we feel that it would be useful to discuss your observations on changes in slope with early maturation in the context of earlier publications. Please, be sure you add text to the discussion addressing these and previously raised issues and caveats regarding human results.

Our answer was focused on the mouse data because only on mouse data we used the 1-40 Hz frequency range to estimate the 1/f slope. For the human data, we used the 1-20 Hz frequency range already from the first iteration of the manuscript. To avoid any ambiguity, we have now explicitly stated this on lines 579-580.

– In terms of the optogenetic data in Figure 4, we feel some additional clarification is required specifically regarding the way data is represented (per trial not per mice), and potential issues of low N.

We modified the text (lines 278-279, 287-288, 292-293, 369-370) to explicitly mention that individual dots correspond to optogenetic stimulation protocols on Figure 4C, Figure 5C and Figure 4—figure supplement 1B-D.

Reviewer #1 (Recommendations for the authors):The authors seem to have addressed my previous comments on an earlier version of this manuscript. I still think their statements regarding the predominance and importance of SST cells are a little strong, and largely unnecessary given they don't study them in this paper, but I guess it is still a matter of debate within the field.

We thank the reviewer for the most helpful feedback and support.

In the manuscript we mention twice the SST+ interneurons (in Introduction and Discussion), referring to published data from other labs. While we agree with the reviewer that our investigation does not address this cell type, we consider important to mention the literature findings that highlight the differences between developing and adult circuits.

In terms of the optogenetic data in Figure 4, their explanation makes sense. It does however seem a little strange to plot 2 trials from the same animal as separate data points. Their analysis seems to account for this, but it does mean that the N is a little low for some time points. That said their new analysis accounting for only the top 50% of active units seems to show a very robust effect consistent with their observations and overall model of inhibition's role in the early network.

To avoid any ambiguity, we modified the main text (lines 278-279, 287-288, 292-293, 369-370) to clearly specify that individual dots correspond to optogenetic stimulation protocols on Figure 4C, Figure 5C and Figure 4—figure supplement 1B-D.

Reviewer #2 (Recommendations for the authors):The authors have appropriately addressed all of my, and the other reviewers', comments. There are still a few typos which I'm sure will be fixed in the final version (e.g. line 707 in the version with tracked changes "such AS a transition in synaptic plasticity rules", line 653 the word In's should be INs (all capital)). Overall, this is a very nice paper that people in the field will enjoy reading.

We thank the reviewer for the most helpful feedback and support. We have corrected the two typos.

[Editors' note: further revisions were suggested prior to acceptance, as described below.]

The manuscript has been improved but there are some remaining issues that need to be addressed, as outlined below:As indicated in the previous decision letter, the editorial consultation on the issue regarding human data agreed to request that this would be addressed directly in the discussion. The decision letter stated "Importantly, we feel that it would be useful to discuss your observations on changes in slope with early maturation in the context of earlier publications. Please, be sure you add text to the discussion addressing these and previously raised issues and caveats regarding human results.". Please, consider this point carefully when providing a revised version. We specifically ask for the issues raised by the non-responding reviewer to be explicitly addressed in the manuscript. Please also note that eLife publishes reviews and decision letters together with manuscripts, so we prefer not to leave important issues unaddressed that were previously raised during reviews and consultation.

As recommended, we added to the manuscript a discussion of our results in the context of previous human studies on early patterns of EEG activity. To the best of our knowledge, there are no previous publications directly investigating developmental changes of the 1/f exponent, beyond the paper by Schaworonkow and Voytek (2021) that we already discuss. Correspondingly, the concern of Reviewer #3 (first revision) refers to the fact that the early EEG is characterized by a robust decline at the lowest frequency range. Our findings do not contradict this observation, but solely point in the direction that the steepness of this decline increases over the two months that are covered by the two datasets that we investigated.

With this, we deem to have addressed all the concerns that have been previously raised during the various rounds of review. As you suggested, we carefully considered the issues raised by the non-responding reviewer and addressed them in the manuscript